# Reconciling discrepant minor sulfur isotope records of the Great Oxidation Event

Benjamin T. Uveges [1] ✉, Gareth Izon [1], Shuhei Ono [1], Nicolas J. Beukes[2,3] & Roger E. Summons [1]

Understanding the timing and trajectory of atmospheric oxygenation remains fundamental to deciphering its causes and consequences. Given its origin in oxygen-free photochemistry, mass-independent sulfur isotope fractionation (S-MIF) is widely accepted as a geochemical fingerprint of an anoxic atmosphere. Nevertheless, S-MIF recycling through oxidative sulfide weathering—commonly termed the crustal memory effect (CME)—potentially decouples the multiple sulfur isotope (MSI) record from coeval atmospheric chemistry. Herein, however, after accounting for unrecognised temporal and spatial biases within the Archaean–early-Palaeoproterozoic MSI record, we demonstrate that the global expression of the CME is barely resolvable; thereby validating S-MIF as a tracer of contemporaneous atmospheric chemistry during Earth's incipient oxygenation. Next, utilising statistical approaches, supported by new MSI data, we show that the reconciliation of adjacent, yet seemingly discrepant, South African MSI records requires that the rare instances of post-2.3-billion-year-old S-MIF are stratigraphically restricted. Accepting others' primary photochemical interpretation, our approach demands that these implied atmospheric dynamics were ephemeral, operating on sub-hundred-thousand-year timescales. Importantly, these apparent atmospheric relapses were fundamentally different from older putative oxygenation episodes, implicating an intermediate, and potentially uniquely feedback-sensitive, Earth system state in the wake of the Great Oxidation Event.

Atmospheric oxygenation represents perhaps the most profound chemical change experienced by the Earth system, revolutionising biogeochemical cycles[1–4] while priming the planet for the rise of complex life[5–9]. Although a wealth of geological observations and geochemical data have been combined to articulate the story of Earth's oxygenation[1,5,10,11] (Fig. 1a, b), the recognition of mass-independent sulfur isotope fractionation (S-MIF) within the geological multiple sulfur isotope (MSI) record remains the most widely accepted means to directly trace the oxygen content of Earth's early atmosphere[12,13].

Manifest as non-zero $\Delta^{3X}$-values ("Methods"; Eqs. 1 and 2)[12,14,15], the generation and geological preservation of S-MIF is linked to low atmospheric oxygen concentrations in several ways: Starting with its production, anoxic $SO_2$ photochemistry remains the only experimentally demonstrated means of creating S-MIF that remotely resembles the geological record[13] (Fig. 1b). Here, in the absence of photon-shielding $O_2/O_3$, photolysis and/or photo(de)excitation reactions are understood to yield reduced (i.e., $S^0$ or $S_8$) and oxidised (i.e., $H_2SO_4$) sulfur phases that carry positive and negative $\Delta^{33}$S values,

[1]Department of Earth, Atmospheric and Planetary Sciences, Massachusetts Institute of Technology, 77 Massachusetts Avenue, Cambridge, MA 02139, USA. [2]DSI-NRF Centre of Excellence for Integrated Mineral and Energy Resource Analysis, Department of Geology, University of Johannesburg, P.O. Box 524, Auckland Park 2006, South Africa. [3]Deceased: Nicolas J. Beukes. ✉e-mail: buveges@mit.edu

**Fig. 1 | Synthesised interpretations of Earth's oxygenation. a** A schematic representation of competing ideas surrounding the oxygenation of Earth's atmosphere[1] contextualised within a framework of biological innovations. **b** The secular evolution of $\Delta^{33}S$ values compiled herein (Supplementary Data File 1). The grey box locates the interval over which atmospheric oxygen is thought to have first accumulated, while the horizontal bars illustrate more traditional redox indicators, whose red and blue colouration discriminates between those disclosing oxic and anoxic conditions, respectively[4,5]. Triangular and circular points represent spot and bulk $\Delta^{33}S$ analyses respectively. Stratigraphic distributions of $\Delta^{33}S$ data from the South African Kaapvaal (**c** core EBA-2; **d** core KEA-4) and Western Australian Pilbara (**e** cores T1–T3) cratons, whose discrepant interpretations have led to the emergence of conflicting models of atmospheric oxygenation. Each of the chronologically constrained lithostratigraphic columns follow those presented with the $\Delta^{33}S$ data[22,25,30,32,33], with the superimposed vertical grey bars illustrating

the ±0.3‰ threshold for identifying S-MIF by $\Delta^{33}S$ alone[25]. The lowermost horizontal blue band in **c**, **d** corresponds to the apparently sustained presence of mass-independent sulfur isotope fractionation (S-MIF) within the Rooihoogte Formation (Ro(oi)) seen throughout the Carletonville area[22,25,30,37]. The subsequent blue horizontal bars in panel **c** mark isolated instances of S-MIF within the Timeball Hill Formation (TBH) that have been interpreted to represent returns to an anoxic atmospheric state (Supplementary Information). Contrastingly, the persistence of low-magnitude non-zero $\Delta^{33}S$ values throughout cores T2–T3 has been ascribed to the crustal memory effect, with atmospheric oxygenation occurring much earlier (blue horizontal bar, **e**, Supplementary Information). Lithostratigraphic abbreviations: Reit. Reitfontain Member, WR Woongarra Rhyolite, BF Boolgeeda Iron Formation, MB1, 2 the diamictites within the Meteorite Bore Member, KF Koolbye Formation, Kaz. Fm. Kazput Formation, BQ Beasley River Quartzite, CB Cheela Springs Basalt.

respectively[12,13,16–18]. Importantly, one-dimensional photochemical models indicate that atmospheric oxygen concentrations must remain below 0.001% of modern levels to retain the photochemical exit channels necessary to communicate S-MIF to the Earth's exogenic sulfur cycle[19]. Further, in terms of preservation, oxygen-deficient environments are less likely to homogenise these photochemical derivatives, thereby increasing their preservation potential and, thus,

their overall likelihood of passing into the sedimentary record[20]. Understandably, it follows that the geological presence of S-MIF is now almost unanimously accepted as a robust geochemical fingerprint of a long-lived oxygen-free atmospheric state, persisting for more than half of Earth's 4.5-billion-year (Ga) history.

Despite the clear and direct links between oxygen and S-MIF, constraining the exact timing and nature of atmospheric oxygenation

has proven difficult. Although it is generally agreed that oxygen began to accumulate within the atmosphere between 2.5–2.3 Ga, different readings of the S-MIF record have been used to portray dramatically different oxygenation trajectories—with some workers arguing for a unidirectional geologically rapid rise in $O_2$[21–23], while others envisage an oscillatory trajectory whereby $pO_2$ repeatedly crossed the threshold necessary to resume S-MIF genesis[24–26] (Fig. 1a–d). While inadequate age constraints and ambiguous stratigraphic correlations[22–24,26–30] currently prevent discrimination between these models (Supplementary Information), both interpretations are reliant on the assumption that sedimentary S-MIF chronicles contemporaneous atmospheric conditions. Again, this premise is not universally accepted and some contend that the operation of a so-called crustal memory effect (CME) decouples the MSI record from coeval atmospheric chemistry[16,31–33]. Specifically, it is argued that the late-Neoarchaean–Palaeoproterozoic MSI record is dominated by S-MIF inherited from an adulterated seawater sulfate reservoir supplied by the oxidative weathering of older crustal pyrite, rather than being a syndepositional atmospheric signal (Fig. 1e). Hence, before we can begin to reconcile the conflicting narratives of atmospheric oxygenation, the feasibility and potential magnitude of a CME must first be addressed.

## Results and discussion

### The crustal memory effect

Noting a post-2.45 Ga shift toward muted positive $\Delta^{33}S$ values, Farquhar and Wing were the first to suggest that weathering-induced crustal recycling of S-MIF may hinder our ability to precisely reconstruct atmospheric oxygenation from the MSI record[16]. Subsequently, predicated on a crustal $^{33}S$ excess[16,18,31,34,35] manifest as a cumulative database mean $\Delta^{33}S$ value of -1.5‰[31], Reinhard et al. presented a series of box-model simulations designed to quantify the CME[31]. Their proof-of-concept approach demonstrated that once mass-independent weathering fluxes permeated an entirely mass-dependant marine sulfur cycle, the weathering-derived non-zero $\Delta^{33}S$ values exceeded those imparted by mass-dependent processes (e.g., >|0.3‰|)[36] for upwards of 10–100 million years[31]. Importantly, such longevity has the potential to induce a temporal lag between the cessation of atmospheric S-MIF production and its loss from the rock record, thereby masking the operation of atmospheric chemistry during this transformative interval of Earth history[31].

While alarming, the paucity of demonstrably contemporaneous MSI datasets has prevented empirical scrutiny of these computational predictions, leaving the significance of the CME untested. Even now, despite a recent explosion of data[21,22,25,30,32,37], opinions surrounding the CME remain divided[21,22,25,26,30,32,33]. For instance, while emerging data from the Western Australian Turee Creek Basin support the notion of a long-lived CME[32,33], purportedly time equivalent (Supplementary Information) observations from the Carletonville area of the South African Transvaal Basin do not (Fig. 1c–e)[22,25,37]—thereby questioning the operation of a CME as a global phenomenon. Furthermore, these spatial differences imply unrecognised biases within the MSI, with the important corollary that an unweighted cumulative mean may ineffectively approximate the isotopic composition of the weatherable sulfur pool, thus confounding previous estimates of the persistence and magnitude of the CME[31,32].

To explore the source of this data–model mismatch, we start by compiling a comprehensive MSI database ("Methods"; Supplementary Data 1), which we then examine in its raw form and, in a reduced state wherein we compensate for potential sample density biases via averaging replicate and intra-sample analyses, termed the Spot Sample Averaged(SSA) database ("Methods"; Supplementary Information). Here, while a cursory evaluation of this new compilation confirms the plateau used by Reinhard et al.[31] to infer the capture of a representative global mean $\Delta^{33}S$ value, augmentation with the data published over the ensuing decade reveals a subtle downward trajectory, resulting in a pre-2.3 Ga mean $\Delta^{33}S$ value of 0.74 ± 1.97‰ (1σ) for the entire database, increasing to 1.12 ± 2.06‰ (1σ) if the SSA database is considered (Fig. 2a).

A more detailed interrogation of the new MSI database reveals lithological, spatial, and temporal biases (Fig. 2; Supplementary Information). Though the lithological bias (e.g., carbonates vs. siliciclastics) was not found to significantly influence our analyses ("Methods"; Supplementary Information; Supplementary Figs. 4 and 5), the impacts of the temporal and spatial biases were substantial. For example, even after compensating for sample replication via the SSA database, more than 70% of the pre-2.3 Ga data were found to originate from either the Kaapvaal or Pilbara cratons, with roughly two-thirds derived from Neoarchaean- (35.5%) and Palaeoproterozoic-aged (33.3%) successions (Fig. 2b, c). To estimate how these biases impact the database mean, we subjected the SSA database to bias-specific bootstrap sampling routines, arriving at two synthetic datasets, which we term the Temporally Adjusted (TA) and Craton Adjusted (CA) datasets ("Methods"). Compared to the mean $\Delta^{33}S$ value obtained from the unweighted SSA database (1.12‰), the mean $\Delta^{33}S$ values obtained from the TA and CA datasets were found to be lower, averaging 0.62 and 0.36‰, respectively (Fig. 2a). Consequently, this simple exercise confirms our data-based inferences, demonstrating that the magnitude of the positive $\Delta^{33}S$ skew apparent within the rock record is, at least partially, a function of bias.

To determine how these biases propagate into our understanding of the size and longevity of the CME, we then incorporated the TA and CA datasets into an updated geochemical model[22] ("Methods"). Augmenting the model with a bootstrap sampling subroutine, the $\Delta^{33}S$ of the seawater sulfate pool was computed utilising the size and isotopic composition of the major marine sulfate fluxes and sinks ("Methods"; Supplementary Information). Here, using the TA dataset, the estimated maximum $\Delta^{33}S$ value of the seawater sulfate reservoir (SWSR) was found to be 0.35 ± 0.2‰ (1σ), which was further reduced to 0.21 ± 0.17‰ (1σ) when utilising the CA dataset (Fig. 3a–b). These bias-corrected estimates of the CME are much smaller than those reported by Reinhard et al.[31] (centred on 1.5‰ and up to 2.5‰) and, indeed, those derived from our updated unadjusted database (0.65 ± 0.6‰; Fig. 3c; "Methods"), thus reaffirming the importance of our bias-correction routine. That said, while small, the fact that the globally integrated CME remains slightly positive despite our bias corrections is a pertinent observation. Inspection of the MSI record shows that the final appreciably negative $\Delta^{33}S$ value occurs at least 60 million years before its positive $\Delta^{33}S$ counterpart (Fig. 1b; Supplementary Information). This asymmetry implicates a muted positive CME persisting beneath an atmosphere capable of S-MIF genesis. Indeed, support for the early onset of a subdued CME is found within a temporal synthesis of molybdenum abundance and isotope data that has similarly been interpreted to signal the onset of oxidative pyrite weathering before $pO_2$ rose sufficiently to curtail S-MIF-yielding photochemistry[38]. These observations, therefore, begin to reconcile conflicting interpretations derived from various atmospheric and marine-based geochemical proxies[30,38], yielding a complementary picture of Earth's oxygenation.

Reminiscent of Reinhard and colleagues' predictions[31], weathering-derived $\Delta^{33}S$ values have long residence times within the SWSR, failing to return to zero within the timespan of our model. Nevertheless, our approach significantly reduces the magnitude of the CME-derived $\Delta^{33}S$ values, arriving at a conservative maximum of approximately 0.5‰, with <0.3‰ being more likely (Fig. 3). This estimate is more consistent with $\Delta^{33}S$ values from Palaeoproterozoic-aged sulfate minerals (|0.1–0.2|‰) that should, in principle, record the coeval SWSR composition[39]. Ergo, beyond validating the MSI record as an archive of Earth's incipient oxygenation, our results can be considered as a conservative discriminatory threshold (i.e., 0.5‰), separating weathering-induced non-zero $\Delta^{33}S$ values from those derived from syndepositional oxygen-free photochemistry.

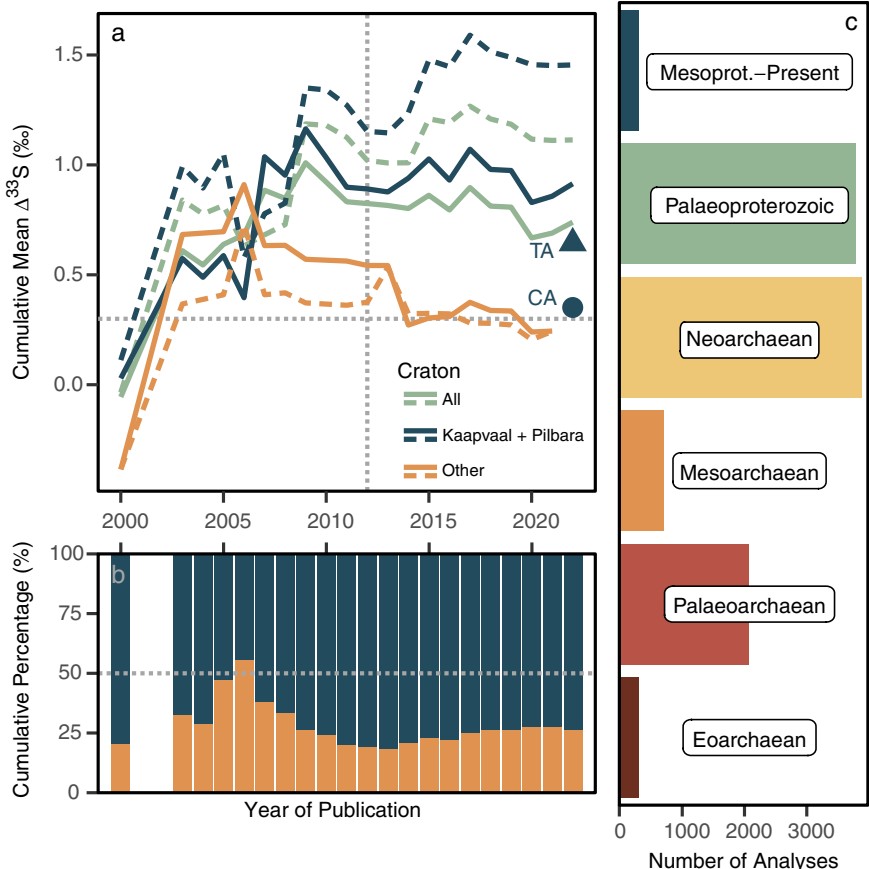

**Fig. 2 | Spatial and temporal biases within the Δ³³S record. a** A cumulative plot depicting the evolving mean Δ³³S value of the pre-2.3 Ga data in the compiled multiple sulfur isotope (MSI) database, with solid and dashed lines representing those derived from the full database and its spot sample averaged (SSA) counterpart, respectively ("Methods"). Line colour distinguishes the craton(s) being analysed, with green signalling the full database, blue illustrating the combined Kaapvaal–Pilbara datasets, and orange showing the remainder. For reference, the respective mean Δ³³S values of the Time Adjusted (TA) and Craton Adjusted (CA) synthetic datasets are plotted as a blue-coloured circle and triangle. A vertical dashed line separates the data considered by Reinhard et al.[31] (i.e., ≤2012) from those published subsequently, while the horizontal dashed line denotes the 0.3‰ Δ³³S-threshold for identifying mass-independent sulfur isotope fractionation (S-MIF)[25]. **b** The cumulative percentage of the Kaapvaal–Pilbara-derived data (blue) relative to those sourced from elsewhere (orange). The horizontal dashed line denotes 50% of the dataset, while the x-axis scaling follows (**a**). **c** A time-binned distribution of Δ³³S data within the unfiltered database. Mesoprot. abbreviates Mesoproterozoic.

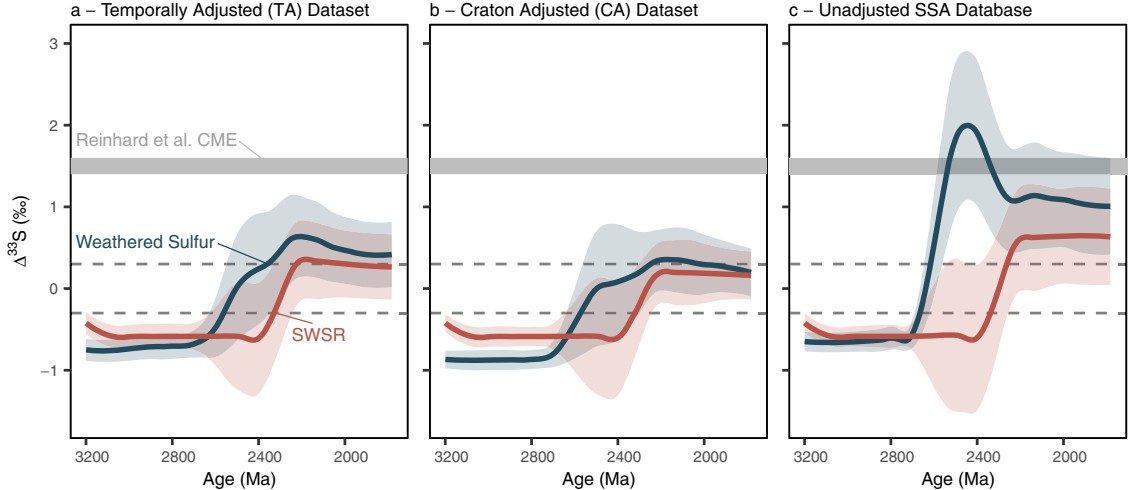

**Fig. 3 | The simulated evolution of the crustal memory effect (CME).** Here, the presented simulations are derived from exploration of **a** the Temporally Adjusted (TA), **b** the Craton Adjusted (CA) and **c** the unadjusted spot sample averaged (SSA) datasets. Colours are common throughout, with blue depicting the Δ³³S value carried by the weathered sulfur flux and orange showing the Δ³³S composition of the seawater sulfate reservoir (SWSR). Colour-coded lines depict the LOESS smoothed mean of the bootstrap outputs, while their envelopes indicate their 1σ uncertainty. The horizontal grey band depicts the existing CME estimate derived using the most feasible boundary conditions[31], while the dashed lines bracket the 0.3‰ Δ³³S-based S-MIF discriminatory threshold.

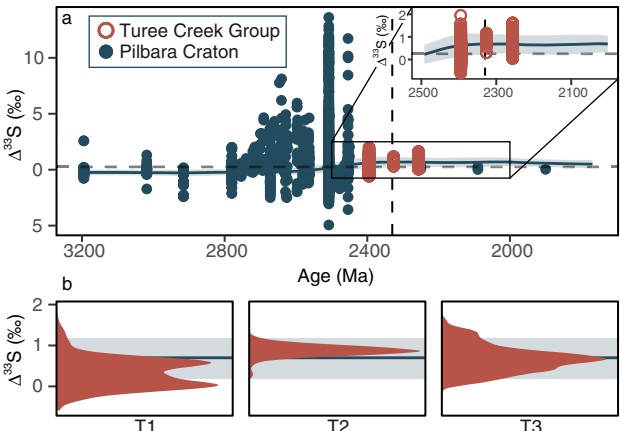

**Fig. 4 | Simulations of the crustal memory effect (CME) within a hypothetical partially hydrographically restricted basin adjacent to the Pilbara Craton.**
**a** Here, the blue line indicates the simulated $\Delta^{33}S$ evolution of the Pilbara-specific weathering flux, with its associated envelope capturing the 1σ uncertainty. Superimposed on these, the available Pilbara-derived $\Delta^{33}S$ data are plotted in blue, while those from the Turee Creek Group are distinguished as red open symbols[32]. The vertical dashed line marks the Great Oxidation Event(GOE) *sensu* Luo et al.[22], while the horizontal dashed line represents the globally expressed CME modelled using the CA dataset (Fig. 3b). The 2500–2000 Ma expansion (insert) follows (**a**). **b** Core-specific density distributions of the $\Delta^{33}S$ data from the Boolgeeda Iron Formation and Turee Creek Group[32], younging from left to right (i.e., cores T1→T3). The horizontal blue line and envelope follow (**a**).

Returning to the MSI record, the small magnitude of the modelled CME, coupled with the fact that less than 5% of $\Delta^{33}S$ values within the entire pre-2.3 Ga database exceed 5‰ (Supplementary Information; Supplementary Fig. 3), strongly supports the interpretation that large magnitude $\Delta^{33}S$ values must originate from oxygen-free photochemistry. Examples of sustained large values are last seen within the Rooihoogte Formation, Transvaal Basin (≤8.76‰), therefore signalling the presence of an oxygen-impoverished atmosphere well after 2.45 Ga[22,25,30] (Supplementary Information).

**Spatial heterogeneities in the expression of the CME**
Independent support for a diminished CME is found within the MSI systematics of contemporary riverine sulfates derived from Archaean-aged terrains[40]. For instance, the circum-zero $\Delta^{33}S$ values (−0.01 ± 0.10‰) found within rivers draining the S-MIF-bearing Superior Craton empirically demonstrate our prediction that catchment-level integration modulates the magnitude of $\Delta^{33}S$ values communicated to the SWSR via weathering. Nevertheless, measurements of sulfates from other riverine systems[41] and, indeed, a South African groundwater sample[40] do, in fact, feature moderate $\Delta^{33}S$ values, demonstrating at least some capacity for the communication of $^{33}S$ anomalies to weathered sulfur fluxes. Accordingly, given the clearly uneven distribution of large $\Delta^{33}S$ values (Fig. 2; Supplementary Table 3, Supplementary Information), a hinterland featuring large $\Delta^{33}S$ values draining into a (semi-)isolated, sulfate-impoverished, basin could hypothetically combine to promote a more pronounced and spatially heterogeneous CME.

Recalling that the Pilbara Craton features some of the most pronounced $\Delta^{33}S$ values seen within the MSI record (Fig. 4; Supplementary Table 3), not to mention the pronounced dichotomy seen between the Australian and South African MSI records (Fig. 1)[25,30,32], we modified our bootstrap sampling subroutine to test whether the apparent persistence of S-MIF within the foreland Turee Creek Basin[26,32] could reflect regional amplification of the CME ("Methods"). Here, drawing from a TA dataset comprising data derived solely from the Pilbara Craton, the resultant weathering flux was found to carry a much larger $\Delta^{33}S$ value

(0.68 ± 0.5‰, 1σ; Fig. 4a) when compared to our estimates of a globally felt CME. Furthermore, the good 1σ agreement between our model output and the $\Delta^{33}S$ data obtained from the two youngest cores (T2 & T3) retrieved during the Turee Creek Drilling Project[32] supports previous calls for a weathering-derived component within the Turee Creek MSI record (Fig. 4a, b)[32,33]. Taking this further, the apparent lithological control seen in the $\delta^{34}S$ and $\Delta^{33}S$ systematics of the Kazput Formation[32,33] implies a higher order depositional control (T3, Fig. 1e). Here, muted $\Delta^{33}S$ values reminiscent of those we estimate for the globally integrated CME are captured within the more distal fine-grained facies, while the near-shore carbonates seem to record more pronounced $\Delta^{33}S$ values akin to those we compute for our Pilbara-derived weathering flux (Figs. 1e, 4). These observations demand that the CME should not be considered in a vacuum and, in reality, the expression of any CME is likely dependent on the interplay between the catchment-specific sulfate weathering flux and sulfate derived from a globally mixed SWSR. This premise is further supported by the fact that barites from the Kazput Formation are distinctly different from SWSR estimates derived from globally-distributed sedimentary sulfates of similar age[39] and, instead, mirror cooccurring sedimentary sulfide $\Delta^{33}S$ values[33] (Fig. 1e). The persistence of S-MIF within Western Australia beyond its demise in South Africa, therefore, need not question the available radiometric age constraints on the Meteorite Bore Member that are used to link the Turee Creek and Transvaal MSI records[26,32,42–44] (Fig. 1e; Supplementary Information).

Contrasting with the monotonous and muted $\Delta^{33}S$ values observed within the Turee Creek Group, several discrete instances of elevated $\Delta^{33}S$ values of up to 2.9‰ are observed to perturb a near-zero baseline within the Timeball Hill Formation, South Africa[25]. While the Kaapvaal Craton is also known to harbour elevated $\Delta^{33}S$ values, application of an analogous craton-specific CME simulation returns a broadly similar weathering flux (0.84 ± 0.6‰), whose theoretical maximum (*c.*1.5‰) is barely half of the most extreme $\Delta^{33}S$ value reported from the Timeball Hill (Figs. 1 c, 5; Supplementary Table 3; Supplementary Fig. 7). Consequently, the pronounced, yet highly variable, nature of these post-2.3 Ga $^{33}S$-enrichments strongly argue against derivation from a regionally amplified CME and, instead, require an alternate explanation.

**Constraining the duration of post-2.3 Ga S-MIF occurrences**
The CME as previously defined[31,32], and adopted herein, does not account for the physical delivery of erosion-derived sulfides. While such detritus has been widely publicised as a potential source of atmospherically estranged sulfur with $\Delta^{33}S$ values of effectively any magnitude[29,45,46], it is hard to envisage a scenario where detrital signals can repeatedly dominate the MSI record without detection−a stance strengthened given the apparent predominance of authigenic pyrites throughout the Carletonville area[22,25,30,37]. Consequently, while we await a detailed and dedicated grain-scale isotopic appraisal designed to unequivocally dismiss a detrital explanation of the elevated Timeball Hill $\Delta^{33}S$ values, it seems rationalisations reliant on recycling can be dismissed. Indeed, Poulton and colleagues[25] argued that the elevated Timeball Hill $\Delta^{33}S$ values recorded primary atmospheric dynamics (Fig. 1c). Enigmatically, however, as testified by a more recent quadruple sulfur isotope appraisal reporting only sub-0.3‰ $\Delta^{33}S$ values within a proximal (<5 km) and indisputably equivalent core (KEA-4), these supposed atmospheric oscillations are not universally resolved[30] (Fig. 1c, d). Consequently, providing the inferred atmospheric significance of the elevated $\Delta^{33}S$ values seen within the Timeball Hill Formation holds true, then the recognised kilometre-scale dichotomy apparent within the Carletonville area requires that the non-zero $\Delta^{33}S$ values must persist over sufficiently stratigraphically restricted intervals to escape detection via conventional sampling campaigns.

To constrain the duration of the Timeball-Hill-housed S-MIF-bearing intervals and, therefore, elucidate their mechanistic driver(s),

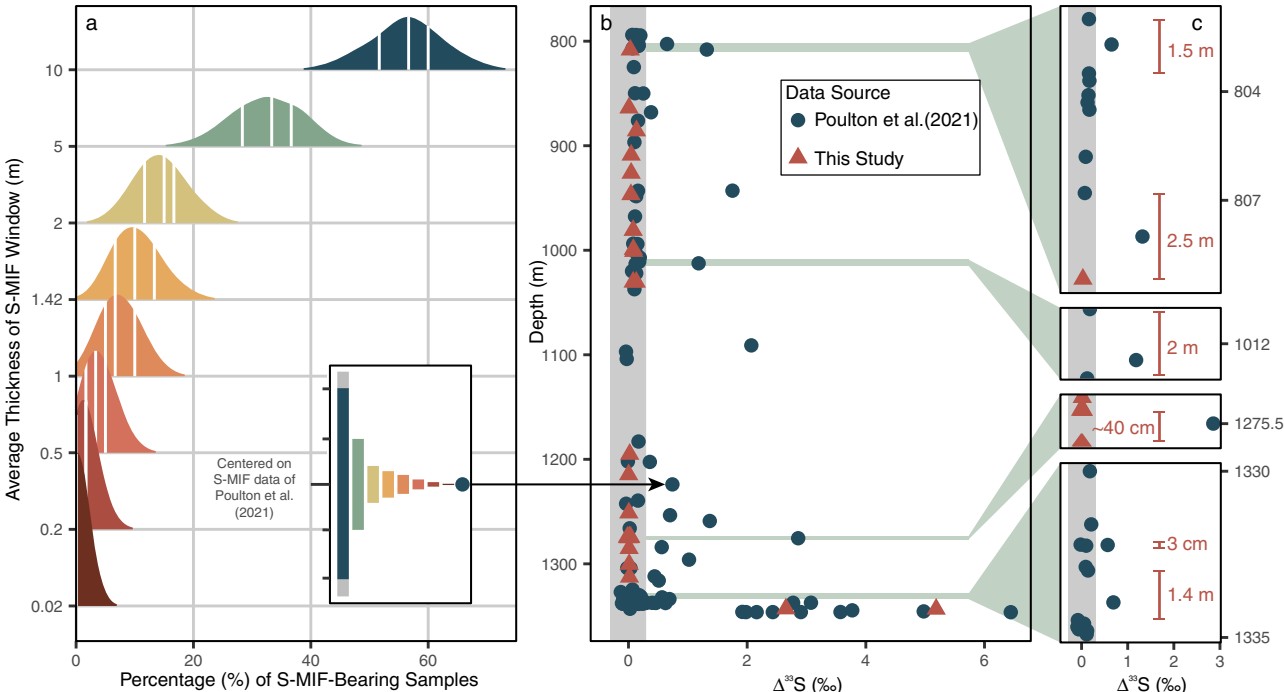

**Fig. 5 | Statistical- and data-based constraints on the longevity of post-Rooihoogte-aged S-MIF-bearing intervals. a** Colour-coded frequency distributions resulting from 1000 replicate 60-sample bootstrap sampling routines, targeting synthetic cores with variably thick mass-independent sulfur isotope fractionation (S-MIF)-bearing windows (y-axis) positioned using the pre-existing EBA-2 $\Delta^{33}$S dataset[25], as exemplified using the sample from 1224.0 m core depth (insert; Supplementary Fig. 10). The median and 25th and 75th quartiles are shown as vertical white lines. **b** Pre-existing $\Delta^{33}$S data from core EBA-2[25] (blue circles), augmented with our new quadruple sulfur isotope(QSI) measurements (red triangles),

are superimposed on the 0.3‰ $\Delta^{33}$S-based S-MIF identification threshold (vertical grey band). The green shading in (**b**) locates the intervals shown in (**c**), providing an expanded view of four stratigraphic intervals where $\Delta^{33}$S values exceeding 0.3‰ were reported[25]. Given the bounding mass-dependant samples, the annotated vertical orange lines denote the maximum thickness of S-MIF-bearing intervals, which, in reality, are likely much smaller; their constraint, however, awaits detailed high-resolution chemical and petrographic conformation. The colour coding follows (**b**).

we started with a simplified Bayesian approach ("Methods"; Supplementary Fig. 9). Here, after converting from the relative abundance of S-MIF in each core, we found the most likely average stratigraphic thickness of each S-MIF re-emergence was 1.42 m, equating to some 0.3–3 million years when combined with average compacted sedimentation rates ("Methods"). Significantly, however, while this Bayesian-derived estimate best explains the available data, Izon and colleagues' inability to detect any S-MIF within their higher-resolution search of the Timeball Hill Formation[30] (Supplementary Table 5) remains salient, emphasising the inherently conservative nature of this estimate ("Methods"). Accordingly, we designed a series of simulations to constrain the likelihood of avoiding S-MIF within a 60-sample sampling campaign akin to that reported by Izon et al.[30] (Fig. 5a, Supplementary Fig. 9; "Methods"; Supplementary Table 5). Strikingly, however, from our 1000 bootstrap replicates, only twice (i.e., 0.2% of the replicates) were we able to avoid detecting S-MIF after prescription of the Bayesian-derived 1.42-metre-thick S-MIF window. Rather, given these boundary conditions, on average, we detected S-MIF 6 ± 2 (1σ) times per replicate, equating to 7–13% of the population size (Fig. 5a; Supplementary Table 6). Extending our statistical approach to increasingly narrower S-MIF windows reveals that their thickness must recede to well below a metre before the likelihood of avoiding S-MIF becomes statistically feasible, implying, in turn, that any re-emergence of S-MIF was likely short lived (Fig. 5a; Supplementary Table 6).

To empirically test the statistically inferred brevity of these S-MIF-bearing intervals, we analysed the MSI systematics of additional samples from core EBA-2, focusing on those bracketing the most pronounced $\Delta^{33}$S value reported from the Timeball Hill Formation (+2.9‰; "Methods"; Fig. 5; Supplementary Data 2)[25]. Here, as predicted, several samples within approximately 30 cm of the S-MIF-bearing sample

found at 1275.5 m core-depth were found to possess mass-dependent MSI systematics, demonstrating that S-MIF was restricted to, at most, 40 centimetres of core (Fig. 5b, c). The presence of several other stratigraphically isolated S-MIF-bearing samples within a metre of those possessing mass-dependant MSI systematics[25] (Fig. 5b, c) supports our inferences and justifies their extension to the wider Timeball Hill Formation. Combining these analytical observations with the preceding statistical analyses reduces the temporal duration of the post-2.33 Ga instances of S-MIF by an order of magnitude, thus signalling the return of S-MIF for perhaps tens-of-thousands of years.

**Implications for the operation of the atmosphere after 2.33 Ga**
Having established that any occurrences of S-MIF in the Timeball Hill Formation must have been short lived, we can now begin to assess the implications for the operation of the atmosphere after 2.3 Ga. First, acknowledging that c.40% of the pre-2.3 Ga $\Delta^{33}$S record falls within 0.3‰ of zero (Supplementary Fig. 8), it could be argued that the elevated $\Delta^{33}$S values seen within the Timeball Hill Formation signal deposition beneath an oxygen-free backdrop. Here, rather than a repeated return to an S-MIF yielding anoxic atmospheric state, this stance requires that the Timeball-Hill-housed S-MIF recurrences record some unknown localised preservation bias, leading to only sporadic expression of elevated $\Delta^{33}$S values. Such an interpretation, however, beyond contradicting the available $\Delta^{36}$S/$\Delta^{33}$S data, is also inconsistent with the wider $\Delta^{33}$S database. For instance, if we exclude the data from the crustally influenced Turee Creek Group, the overwhelming majority (>90%) of the 2.3–2.1 Ga $\Delta^{33}$S record comprises values between −0.3 and 0.3‰ (Supplementary Fig. 8). Thus, rather than mirroring the older MSI record derived from an almost unanimously accepted anoxic background state, the Timeball-Hill-derived

$\Delta^{33}$S distribution is much more reminiscent of the post-GOE record. Within this framework, the loss of S-MIF within the Rooihoogte Formation appears to mark a fundamental transition within the Earth system, with the global predominance of S-MIF seen prior to the Timeball Hill Formation (Supplementary Fig. 8) signalling a dominantly anoxic background state that yields to a more oxidising equivalent at around 2.3 Ga[22,30].

Accepting the premise that the post-2.3 Ga atmosphere was predominantly oxygenated enough to inhibit S-MIF production/preservation, we are left with describing a mechanism for the short-lived re-emergences of S-MIF in the Timeball Hill Formation. Given that stratospheric volcanic eruptions are known to communicate $\Delta^{33}$S values as large as |2.0|‰ to contemporary ice-core records[47–49], such events could hypothetically act as a geologically instantaneous source of S-MIF in more ancient settings. Indeed, the sulfate-impoverished oceans of the Palaeoproterozoic[22,45,50,51], in principle, would offer little buffering capacity against such a point-source injection of S-MIF. That said, while S-MIF genesis and preservation against a modern oxygenated backdrop is a good starting point, its application to the Timeball Hill Formation and, indeed, the wider sedimentary record is problematic. Aside from the distinct differences between the ice-core (−4.3)[49] and rock-housed $\Delta^{36}$S/$\Delta^{33}$S records (−0.9)[14], their mode of preservation is also fundamentally different. In essence, the survival of ice-core-housed $\Delta^{33}$S anomalies is not only a function of evading dilution with mass-dependently fractionated sulfur, but it specifically requires time-resolved differential rainout. Here, rather than spatially separating two isotopically distinct sulfur phases (i.e., $S_8$ and $SO_4^{2-}$), the ice-core housed S-MIF bearing sulfates are separated in time over the course of several years by the relatively rapid and pristine accumulation of the snow/ice pack[47–49]. In the absence of this time separation, mass-balance requires that $^{33}$S anomalies would be eradicated when the two pools were re-mixed[47]. Given the marine setting of the Timeball Hill Formation, it is realistic to assume that any volcanogenic sulfates would be homogenised, and therefore any $^{33}$S anomaly would be erased by mixing within the SWSR and the subsequent diagenetic realm. Theoretically, these issues could be circumnavigated by the model proposed by Gallagher et al., who envisage the volcanic injection of subduction-derived recycled sedimentary sulfur carrying positive $\Delta^{33}$S values[52]. While we concede that such a model offers an intriguing, not to mention potentially substantial, point-source of S-MIF, recalling the logic presented previously for the CME, we dispute its relevance in this case. Rather, we anticipate that, similar to the mixing that occurs under surficial weathering, homogenisation within the subduction system would minimise the $\Delta^{33}$S value of the re-entrained sulfur pool to values smaller than those seen in the Timeball Hill Formation (Supplementary Information, Supplementary Table 4.1–4.3).

All told, we are unable to explain the transient post-2.3 Ga instances of S-MIF without invoking high-order atmospheric dynamics with episodic relapses toward an anoxic atmospheric state. While we stress that the pre-Rooihoogte-aged oxidation episodes await definitive demonstration (Supplementary Information), it is clear that the S-MIF variability seen within the Timeball Hill Formation implicates atmospheric dynamics operating on fundamentally different timescales than their alleged precursors[24–26]. Such a system not only requires a background state susceptible to transient atmospheric perturbations, but also one that is conducive to S-MIF preservation. Preservation of significantly positive S-MIF signatures is thought to require a small SWSR[53], which implies slow rates of oxidative weathering of continental sulfides and, by extension, relatively low $p$O$_2$. It follows, therefore, that the periodic reappearance of S-MIF in the Timeball Hill Formation may signal a protracted, intermediate, and extremely sensitive atmospheric state that was uniquely susceptible to perturbation as oxygen contents vacillated around the threshold for S-MIF genesis and preservation. Speculatively, such a state could have been established and maintained through the interplay of biological feedbacks[54,55] encountered as organisms gradually evolved the biochemical machinery to thrive in increasingly more oxidising regimes[5]. Superimposed on this intermediary atmospheric state, large injections of reducing gasses, perhaps sourced via the emplacement of large igneous provinces or, more likely, through climate-paced methane fluxes, could have episodically outpaced the biological O$_2$ flux to ephemerally reinstate S-MIF production[56]. Interestingly, support for these claims can be found in recent photochemical modelling efforts that demonstrate the feasibility of 10–100 thousand-year returns to oxygen-free S-MIF yielding atmospheres, providing the predominant boundary conditions remain near to those necessary for S-MIF genesis[56]. The prediction of a highly dynamic and intermediate atmospheric state represents a paradigm shift, departing from the conventional view that atmospheric oxygenation occurred as a step function between bistable endmember compositions[57,58]. As such, there is now a pressing need for more detailed high-resolution MSI studies to better constrain the nature and duration of this transitory period, and thereby refine the framework of Earth's most significant redox revolution.

## Methods

### Isotopic nomenclature

Following convention, sulfur isotope data are expressed in delta ($\delta$) notation, reflecting permille (‰) deviations of the less abundant isotope, $X$ ($^{33,34,36}$S), normalised to $^{32}$S, relative to the same ratio in the international reference standard, Vienna Canyon Diablo Troilite, (VCDT):

$$\delta^{3X}S = [(^{3X}S/^{32}S)_{\text{sample}}/(^{3X}S/^{32}S)_{\text{VCDT}}-1]\times 1000 \qquad (1)$$

Most processes fractionate S-isotopes mass-dependently, whereby $\delta^{3X}S \approx \lambda_{3X}\times\delta^{34}S$, with respective $\lambda_{3X}$ values of 0.515 and 1.91 for $\delta^{33}$S and $\delta^{36}$S. Departure from mass-dependent behaviour, termed mass-independent fractionation (MIF), however, is expressed in capital-delta ($\Delta$) notation, where:

$$\Delta^{3X}S = 1000\times[\ln(\delta^{3X}S/1000+1)-\lambda_{3X}\times\ln(\delta^{34}S/1000+1)] \qquad (2)$$

### Database description

The sulfur isotope data used herein ($n = 10{,}765$ with at least $\Delta^{33}$S values) were compiled from 84 studies, forming a new and comprehensive database complete with metadata (Supplementary Information; Table 1; Supplementary Data 1). Given that many studies fail to report $\delta^{33}$S and $\delta^{36}$S values, to assess the effects of non-linear mixing in $\Delta$–$\Delta$ space[14,36] we calculated the missing $\delta$-values via algebraic rearrangement of Eqs. 1 and 2. Overall, however, our presented solutions were found to be broadly equivalent irrespective of whether the mean $\Delta^{33}$S values were used directly or whether the $\Delta^{33}$S values were calculated retrospectively. To remove complications arising from the myriad of published sample lithological descriptions, our assessment of lithological bias uses a simplified parameter (simp.lith), allowing the compiled data to be allotted to one of ten generalised lithotypes (i.e., high energy siliciclastic, low energy siliciclastic, carbonates, sulfates, chemical, microbial, altered/late addition, glacial/diamictite/detrital, igneous/volcanic sed., and none provided). The individual lithologies assigned to each of our generalised lithotypes are listed in Supplementary Table 6. Finally, to ensure that inclusion of replicate sample analyses did not impact our analyses and resultant conclusions, we compared the full database to a Spot Sample Averaged database (SSA) where all stratigraphic replicates and intra-sample measurements (e.g., SIMS, KrF spot fluorination, in-situ laser, SHRIMP, CO$_2$ spot laser fluorination, LA MC-ICP-MS) were averaged. All database interrogation, statistical analysis and figure generation was performed in R via

**Table 1 | Description of data and metadata included in multiple sulfur isotope (MSI) database**

| Database header | Description |
|---|---|
| Craton | The craton from which the sample is derived |
| Supergroup | The supergroup from which sample is derived |
| Group | The group from which the sample is derived |
| Subgroup | The subgroup from which the sample is derived |
| Formation | The formation from which the sample is derived |
| Member | The member from which the sample is derived |
| Core | The core from which the sample is derived. If sampled from outcrop, listed as 'Outcrop' |
| age.max | Maximum published nearest age for a given sample |
| age.min | Minimum published nearest age for a given sample |
| age.mean | Mean of age.min and age.max |
| age.source | Source(s) for age.min and age.max |
| Sample.ID | Sample identifier. If not provided, the ID was constructed in the form of "Core-Depth" |
| Replicate | Column with a unique identifier for replicate analyses of the same sample. This includes individual spots from the same bulk sample |
| Lithology | Reported lithology |
| Mineralogy | Reported mineralogy. In general, for bulk samples, unless explicitly stated within the source publication, this was left blank |
| Phase | Phase of sulfur measured (i.e., sulfide, sulfate, sulfide + organic S, organic S, total S) |
| core.rim | For spot measurements, denotes whether the spot sampled the core or rim of the mineral analysed. Not always specified |
| Depth | Metres core-depth where the sample was taken. If sample was from an outcrop, a stratigraphic height is given, where applicable |
| d34s | $\delta^{34}$S (‰) |
| sig.d34s | Reported uncertainty in $\delta^{34}$S (‰) |
| D33S | $\Delta^{33}$S (‰) |
| sig.D33S | Reported uncertainty in $\Delta^{33}$S (‰) |
| D36S | $\Delta^{36}$S (‰) |
| sig.D36S | Reported uncertainty in $\Delta^{36}$S (‰) |
| Analysis.Type | Type of analysis (e.g., bulk, $SF_6$; SIMS; EA-CF-IRMS etc.) |
| Chemistry | Preparative chemistry performed prior to analysis |
| Source | Source of the data |
| DOI | Digital Object Identifier of published data source |
| Date | Year of publication |

**Table 2 | Description of variables used in Eqs. 4–9**

| Variable | Description |
|---|---|
| $S$ | Seawater sulfate reservoir (SWSR) at time = $t_{step}$ |
| $S_i$ | SWSR at time = $t_{step} - dt$ |
| $F_{VSulf}$ | Flux of volcanic sulfur into the SWSR in the form of sulfate |
| $F_{VS8}$ | Flux of volcanic sulfur into the SWSR the form of $S_8$ |
| $F_{ws}$ | Flux of weathered sulfur into the SWSR |
| $F_{bpy}$ | Flux of buried pyrite out of the SWSR |
| $F_{bgyp}$ | Flux of buried gypsum out of the SWSR |
| $F_{S8py}$ | Flux of $S_8$ directly into the buried pyrite phase |
| $D_{Sulf}$ | $\Delta^{33}$S of the SWSR |
| $D_{VSulf}$ | $\Delta^{33}$S of volcanic sulfur into the SWSR in the form of sulfate |
| $D_{VS8}$ | $\Delta^{33}$S of volcanic sulfur into the SWSR the form of $S_8$ |
| $D_{ws}$ | $\Delta^{33}$S of weathered sulfur into the SWSR |
| $D_{bpy-s}$ | $\Delta^{33}$S of buried pyrite out of the SWSR |
| $D_{bpy}$ | $\Delta^{33}$S of buried pyrite including the direct incorporation of $S_8$ |
| $D_{bgyp}$ | $\Delta^{33}$S of buried gypsum out of the SWSR |

negative counterpart to the seawater sulfate reservoir.

$$Ox = \frac{O_2{}^{0.5}}{O_2{}^{0.5} + K_{wpy}{}^{0.5}} \quad (3)$$

$$D_{bpy-s} = D_{Sulf} \quad (4)$$

$$D_{bgyp} = D_{Sulf} \quad (5)$$

$$S = S_i + F_{VSulf} + F_{VS8} + F_{ws} - F_{bpy} - F_{bgyp} \quad (6)$$

$$D_{Sulf} = \frac{D_{Sulf} \times S_i + D_{VSulf} \times F_{VSulf} + D_{VS8} \times F_{VS8} + D_{ws} \times F_{ws} - D_{bpy-s} \times F_{bpy} - D_{bgyp} \times F_{bgyp}}{S} \quad (7)$$

$$D_{bpy} = \frac{D_{Sulf} \times F_{bpy} + D_{VS8} \times F_{S8py}}{F_{bpy} + F_{S8py}} \quad (8)$$

$$D_{ws} = \text{Output from bootstrap sampling subroutine} \quad (9)$$

$$\text{Time Weight} = e^{-0.001 \times (t_{step} - t - 1)} \quad (10)$$

the Tidyverse family of packages[59], as well as the patchwork[60], ggridges[61] and MetBrewer[62] packages.

Besides forming the backbone of our statistical analyses, the compiled database has been used to create an interactive HTML interface that allows the construction of user-defined time *vs.* $\Delta^{33}$S plots (Supplementary Information). The HTML interface was generated using R and the Plotly package[63].

**Modelling the crustal memory effect**

The base geochemical model was run in MATLAB. The model follows ref. [22]., however, we alter the oxidative pyrite weathering constant to emulate empirically demonstrated half-order kinetics[64] (Eq. 3; Table 2). The sulfur isotope model was constructed using the flux outputs of the base geochemical model, including oxygen concentrations (Ox) and Eqs. 4–10 (Table 2). Here, the isotopic imbalance between the various sulfur reservoirs was achieved via the direct sequestration of a given proportion of $S_8$ ($F_{S8py}$) within the sedimentary pyrite pool, communicating a positive $\Delta^{33}$S value to the continental reservoir and its

The $\Delta^{33}$S value of the weathered sulfur flux at each time step ($t_{step}$) was calculated directly from the database using a bootstrap sampling subroutine. Here, two different versions of the database were analysed, with each optimised to minimise biases: First, we generated a Temporally Adjusted (TA) synthetic dataset whose overall $\Delta^{33}$S distribution reflected that of the precursor SSA database, yet featured a consistent sample density within each 50-million-year time-bin. Next, we derived a second Craton Adjusted (CA) synthetic dataset, using the same parameters as before, however, now we stipulated that once featured within a given time-bin each craton was equally represented. For the model run using the unadjusted database, the raw form of the SSA database was utilised using the same model parameters as above. Naturally, to protect the integrity of our findings, samples that were initially described as altered and/or late addition were excluded from our analyses.

The $\Delta^{33}$S value of the weathered sulfur flux at each time step ($t_{step}$), was calculated directly from the TA and CA synthetic datasets using a MATLAB-based bootstrap sampling subroutine. Here, each calculation was restricted to older samples (i.e., $t > t_{step}$) whose likelihood of selection was reduced as $t$ and $t_{step}$ diverged, via an age weighting equation (Eq. 10) modelled after Ref. [65]. Importantly, an oxygen scalar (Ox; Eq. 3) designed to curb sulfide weathering at low $pO_2$ was also employed, allowing sulfide weathering to become more pronounced as atmospheric oxygen rose. Using these weighting protocols to quasi-regulate the age and sulfur phase of selection, the bootstrap sampling subroutine then selected $n$ samples and calculated their mean. Repeating this process $m$ times then allowed us to assess the variability of the output. For instance, considering Fig. 3, the lighter coloured envelopes depict the standard deviation ($\pm 1\sigma$) of the bootstrap means, while the central line gives the mean of the replicate bootstrap outputs (i.e., the grand mean). After exploring the sensitivity of the model to these parameters, respective $n$ and $m$ values of 10 and 20 were selected for the final model (Supplementary Information; Supplementary Fig. 6).

Given its targeting of the same sulfide pool as oxidative weathering, photochemically catalysed pyrite oxidation, as recently described by Hao et al.[66], would not significantly change our isotopic estimates of the post-2.3 Ga CME and, therefore, is not considered in the model. This process, we concede, could provide an under-appreciated source of sulfate to the oceans that has the potential to drive the $\Delta^{33}$S composition of the SWSR closer to 0‰ prior to the intensification of oxidative sulfide weathering.

## Statistical analyses of post-Rooihoogte-aged S-MIF

To better characterise the prevalence of S-MIF after 2.33 Ga within the Transvaal Basin, and thus ascertain its regional–global significance, we employed a simple Bayesian approach coupled with a data-parameterised[25,30] bootstrap-driven synthetic sampling experiment (Fig. 5; Supplementary Figs. 9 and 10). Here, to avoid potential sampling density biases introduced by the presence of variably sized and distributed diabase intrusions, we focused our attention on the intrusion-free lower Timeball Hill Formation prior to the Gatsrand Member (Fig. 2; Supplementary Fig. 10). In each of the explored cores, the expression of the lower Timeball Hill Formation is roughly equivalent, spanning 168.5, 151.6, and 158.3 m in cores EBA-1, EBA-2, and KEA-4, respectively[22,25,30,37]. Comparison between these different expressions of the lower Timeball Hill Formation is presented in Supplementary Table 5.

Using the parameters outlined in Supplementary Table 5, we started with a simplistic Bayesian approach to estimate the likelihood that a given stratigraphic interval is S-MIF bearing. In essence, we can liken the mass-dependency preserved within the lower Timeball Hill MSI record to a biased coin flip, where S-MIF-bearing samples (i.e., $\Delta^{33}$S ≥ 0.3) are assigned heads and their mass-dependant counterparts (i.e., $\Delta^{33}$S ≤ 0.3) assigned tails. Here, our hypothetical coin is biased, meaning that the odds of returning heads or tails is unfairly weighted (i.e., ≠50:50). Consequently, informed by the outcomes of a given number of coin flips, we are effectively determining the bias ($\theta$) of this hypothetical coin. In this context, each coin flip equates to an analytical $\Delta^{33}$S measurement and $\theta$ to the likelihood of randomly selecting a S-MIF-bearing sample from a core.

Bayesian analysis requires the definition of a prior distribution and a likelihood distribution. Accordingly, applying Poulton and colleagues' data[25] to Eqs. 11 and 12, a prior was generated using a beta distribution:

$$p(\theta|a,b) = \frac{\theta^{a-1} \times (1-\theta)^{b-1}}{B(a,b)} \quad (11)$$

$$B(a,b) = \int_0^1 d\theta\, \theta^{a-1} \times (1-\theta)^{b-1} \quad (12)$$

This prior describes the probability of a particular $\theta$, given the parameters $a$ and $b$, or p($\theta|a,b$), which represents the respective number of 'heads' (S-MIF) and 'tails' (S-MDF) in cores EBA-1 and EBA-2. Here, the mean $\theta$ of the resultant distribution was 0.203 and its mode was 0.190 (Supplementary Fig. 9a), empirically consistent with the 20.3% (=13/64) S-MIF detection frequency reported by Poulton et al.[25]

Next, a Bernoulli likelihood distribution was generated using the data presented by Izon et al.[30] (Eq. 13):

$$p(D|\theta) = \theta^z \times (1-\theta)^{N-z} \quad (13)$$

with $N$ representing the number of samples in the lower Timeball Hill Formation, and $z$ being the number of S-MIF-bearing samples (60 and 0, respectively; Supplementary Fig. 9b). This distribution describes the likelihood of reproducing the data ($D$) at a given $\theta$, or $p(D|\theta)$. Returning to our numismatic analogy, Eq. 13 gives the likelihood of returning 'tails' 60 times given a specific weighting of a coin. Naturally, given Izon and colleagues' 0% S-MIF detection frequency in the lower Timeball Hill Formation within core KEA-4[30], the mode of the resultant Bernoulli likelihood distribution was 0.

Finally, by combining the previously described prior (i.e., $p(\theta|a,b)$) and likelihood (i.e., $p(D|\theta)$) distributions a posterior distribution was constructed using Bayes theorem (Eqs. 14 and 15; Supplementary Fig. 9c):

$$p(\theta|z,N) = \frac{\theta^{z+a-1} \times (1-\theta)^{N-z+b-1}}{B(z+a,N-z+b)} \quad (14)$$

$$B(Z+a,N-z+b) = \int_0^1 d\theta\, \theta^{z+a-1} \times (1-\theta)^{N-z+b-1} \quad (15)$$

This final distribution describes the updated probability of $\theta$ given the new data parameters $z$ and $N$, or, p($\theta|z,N$). The resultant mean of the posterior distribution was 0.105 and its mode was 0.100. Given the combined probabilities in the posterior distribution, the most likely $\theta$, or probability of randomly selecting an S-MIF-bearing sample (i.e., the coin returning 'heads') is 10.0–10.5%. This likelihood can then be translated into the proportion of a given core possessing mass-independent versus mass-dependant S-isotope systematics. Returning to the proximal (<5 km) Carletonville cores, given that the lower Timeball Hill Formation is roughly of equivalent thickness, we assume that the relative occurrence of S-MIF and the stratigraphic interval over which it's expressed would be broadly analogous within each core. Maintaining these assumptions, we predict that 10.25% of the lower Timeball Hill Formation within core EBA-2 (151.6 m), equating to 15.54 m, should be S-MIF-bearing, which, if accommodated by the 11 S-MIF-bearing samples found in EBA-2[25], results in an average S-MIF-bearing interval of 1.42 m.

Pursuing this tactic further, we devised a bootstrap sampling method that would allow us to estimate the likelihood of failing to find S-MIF within a 60-sample sample set taken at random from synthetic cores constructed using different stratigraphic distributions of S-MIF. Designed to emulate the typical sample thickness used by Izon et al.[30] and, indeed, that typically administered by core repositories, we constructed a 150-m-long synthetic core comprising 3000 5-cm-thick samples. Then, by centring a hypothetical S-MIF window of a given thickness (0.02–10.0 m) on the S-MIF-bearing samples (i.e., $\Delta^{33}$S > 0.3‰) reported by Poulton et al.[25], we prescribed the mass-dependency of the surrounding samples (Supplementary Fig. 10). Selecting 60 samples from the resultant synthetic core, and repeating the operation 1000 times, then yielded a frequency distribution illustrating how often S-MIF would be expected to be detected within a hypothetical 60-sample analytical campaign. Repeated analysis with variably thick S-MIF windows (0.02–10.0 m), demonstrated that non-

detection of S-MIF within a 60-sample population only became likely as the size of the S-MIF window diminished (Fig. 5a; Supplementary Table 6). Recalling our Bayesian analysis, the likelihood of randomly selecting 0 or 11 S-MIF-bearing samples is roughly equivalent given a 1.42-m-thick S-MIF window. That said, it is important to stress that the likelihood of achieving either is very low, falling outside of 2σ of the mean (Fig. 5a). It follows, therefore, that while our Bayesian approach yields a S-MIF window that most adequately explains the current data, it is unlikely to accurately describe the true distribution of S-MIF. Indeed, it is only when the thickness of the prescribed S-MIF window drops to well below a metre do the chances of avoiding S-MIF within a 60-sample analytical campaign become probable (Fig. 5a; Supplementary Table 6).

## Sulfur isotope analysis

Augmenting previously published EBA-2 data[22], herein we report an additional 32 quadruple sulfur isotope (QSI) measurements. First, to ensure consistency, we targeted two previously reported samples from the Rooihoogte Formation[22] before measuring a new sample suite spanning the Timeball Hill Formation (Supplementary Data 2). While previous work over-looked acid volatile sulfur (AVS)[22], herein we follow the sequential two-step approach detailed in Izon et al.[30] that liberates AVS upon reaction with ethanoic 6 M HCl followed by the release of chromium reducible sulfur (CRS; principally pyrite) via reduction using acidified 2 M $CrCl_2$. In both AVS- and CRS-yielding refluxes, the resultant $H_2S$ was swept into zinc acetate traps where it was captured as zinc sulfide before being converted to silver sulfide ($Ag_2S$) with silver nitrate. Given the relative paucity of QSI data within core EBA-2, we universally converted the $Ag_2S$ to sulfur hexafluoride ($SF_6$) via overnight reaction with excess of fluorine gas ($F_2$) at 300 °C. The resultant $SF_6$ was then purified cryogenically at liquid-nitrogen-temperatures before final isolation by preparative gas chromatography within the Geobiology laboratory at MIT. The QSI isotope composition of the pure $SF_6$ was measured by dual-inlet gas-source isotope ratio mass spectrometry using a Thermo-Finnigan MAT 253 equipped with four collectors arranged to measure the intensity of $SF_5^+$ ion beams at mass/charge ratios ($m/z$) of 127, 128, 129 and 131 ($^{32}SF_5^+$, $^{33}SF_5^+$, $^{34}SF_5^+$, and $^{36}SF_5^+$). Over the course of this study, several fluorinations of the IAEA-distributed reference materials S1, -S2 and -S3 were performed, returning 1σ $δ^{34}S$, $Δ^{33}S$ and $Δ^{36}S$ uncertainties of better than 0.4, 0.022 and 0.16, respectively (Supplementary Data File Table SX2). Here, the resultant VCDT-normalised values are inseparable from their certified values and, indeed, those reported initially by Ono et al.[67] (Table SX2). Importantly, the reproducibility of these pure $Ag_2S$ standards approximates those calculated from repeated sample processing (i.e., extraction–fluorination; Supplementary Data 2).

## Data availability

The compiled database and the MSI data generated in this study are provided in Supplementary Data 1 and 2 respectively with an inter-active HTML interface plot of the geological $Δ^{33}S$ record through time provided as Supplementary Data 3.

## Code availability

The code used in this study can be found online in a GitHub repository https://github.com/buveges/Sulfur-MSI-Database along with an updating version of the MSI database. The MATLAB model was originally created by S.O. for Luo et al.[22], and subsequently modified and augmented by S.O. and B.T.U. for this study. The R code was written by B.T.U.

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

## Acknowledgements

We dedicate this paper to our co-author Prof. Nicolas J. Beukes who sadly passed away after the acceptance of this contribution. As a colleague, mentor, and friend "Prof. Nic" worked tirelessly to bring South African geology to the world. Beyond his many contributions, Nic's work has, and continues to, revolutionise our understanding of Earth's evolution. This study was supported by the Simons Collaboration on the Origins of Life (#290361FY18 to R.E.S. & #874698 to B.T.U. and R.E.S.) and the National Science Foundation (#EAR-1338810 to R.E.S. and S.O.). G.I. acknowledges receipt of a MISTI Global Seed Award. N.J.B. acknowledges financial support from DSI-NRF CIMERA in South Africa. We recognise formative discussions with various members of the Summons Lab.

## Author contributions

B.T.U. and G.I. designed the study. B.T.U. compiled the chemical data, constructed the database and performed its statistical interrogation. B.T.U. implemented and augmented the geochemical model with guidance from S.O. N.J.B. and S.O. collected the samples from which G.I. conducted the isotopic analyses. B.T.U. and G.I. synthesised the data–model interpretations, writing the manuscript with input from all authors. R.E.S., S.O., G.I., N.J.B., and B.T.U. acquired the funding to support various aspects of this study.

## Competing interests

The authors declare no competing interests.
