## [Peer Review File · Nature Communications]

Reconciling discrepant minor sulphur isotope records of the Great Oxidation EventReviewer #1 (Remarks to the Author):

This study uses statistical tools and modeling to test the operation of a previously proposed global scale crustal memory effect (CME) of sedimentary recycling of sulfur isotope anomalies within the early Paleoproterozoic record of the Great Oxidation Event (GOE), ca. 2.5-2.3 billion years ago. It is a needed state of the art assessment of the efficacy of the mass-independent fraction sulfur isotope (S-MIF) record for tracking global signatures of the transition to an oxygenated atmosphere of greater than 10^{-5} PAL O₂. The results of the test of the CME are convincing, showing that the CME should not leave a globally resolvable sulfur isotope signature. While this study is worthy and compelling, there are some aspects that might deserve a further look, as suggested below.

Discrepancies between 2.45-2.2 Ga sulfur isotope records from the Turee Creek Basin (Pilbara craton) versus Transvaal Basin, and within sections from the Carletonville area in the Transvaal Basin (Kaapvaal craton) speak to the significant spatial sensitivity of the sulfur isotope record in the aftermath of the initial rise of oxygen ca. 2.45 Ga. After all, the records we are left are from variably restricted basins. We do not know what records might be considered representative of the global ocean versus more localized signals within restricted basins. A local CME, instead of only a global CME, may need to be tested to fully lay this issue to rest. Could the model of the CME be pushed further, to apply to the scale of a basin instead of the whole, global, seawater sulfate reservoir?

In another example of spatial variability, although the cited (ref. 38, lines 108-113) Torres et al., 2018 study is a clever one, S-MIF in contemporary sulfates in surface waters from Archean-aged terrains has been measured from S. Africa (Asael et al., 2017, Sulfur non mass dependent anomalies in modern river water of Archean catchment, Goldschmidt abstract). Both of those results are correct (though, taken individually, their interpretations may differ): weathering of Archean terrains today does and does not contribute S-MIF-bearing sulfate to rivers, likely depending on different catchment-specific factors. This could be even further evidence that the nature of sedimentary recycling of S-MIF (or, the crustal memory effect), if it shows up in the rock record, should be inherently localized.

Finally, S-MIF can certainly be produced in an oxygenated atmosphere, with such records from as recent as 1991 from volcanic eruptions are found preserved in ice core (Savarino et al., 2003), provided such more recent volcanic sulfate does differ in $\Delta^{36}\text{S}/\Delta^{33}\text{S}$ systematics versus Archean age sulfur. Nonetheless, it is their preservation that is critical to S-MIF records. As described, the key to S-MIF preservation is that the mixed phases of S-MIF bearing sulfur can be spatially separated and not homogenized at the Earth's surface. It could be that the analogy made between anthropogenic point-sources of atmospheric lead and the rain-out of this lead is indeed like the production and preservation of ancient S-MIF (Gallagher et al., 2017). Depending on the sensitivity of a particular Paleoproterozoic basin (due to sulfate concentration, sulfate weathering influx, facies controls, and biological controls), point sources of volcanogenic sulfate that do not have much long-lived global significance could make an outsize impact on S-MIF records from individual basins.

Some specific comments:

General comment on the figures: the use of solid, filled, symbols makes it difficult to appreciate the sample density on the plots. Could unfilled symbols be used instead?

L114-116: Inspection of the $\Delta^{33}\text{S}$ record shows that the last appreciably negative value occurs well before its positive $\Delta^{33}\text{S}$ counterpart (Fig. 1B).

It would be nice to give the respective dates and S isotope values of the last appreciably negative and positive $\Delta^{33}\text{S}$.

L171-176: Consequently, provisionally accepting previous interpretations that the elevated $\Delta 33S$ values seen within the Timeball Hill Formation (Fig. 1C) do in fact faithfully record atmospheric chemistry²⁵, we ask the question how can we reconcile the apparently discrepant Carletonville MSI records? The most parsimonious explanation is that these intervals must be stratigraphically isolated and therefore represent geologically brief returns to an oxygen-free atmospheric state capable of S-MIF genesis and export²⁵.

If the occurrence of S-MIF in Timeball Hill is indeed a contemporaneous atmospheric signal, does this really require an oxygen-free atmosphere? S-MIF can certainly be produced in an oxygenated atmosphere, however its preservation is another challenge. For example, the 1991 Pinatubo volcanic eruption resulted in sulfur with elevated $\Delta 33S$ preserved in ice core (Savarino et al., 2003). The caveat is that the $\Delta 36S/\Delta 33S$ isotope systematics are different in the ice core record of the Pinatubo eruption compared to Archean $\Delta 36S/\Delta 33S$ signatures, whereas the former gave a steeper slope (though, interestingly, at -4.3 this MIF slope is close to the "MDF" slope of -3.2 identified by Poulton et al., 2021 in Rooihogte-Timeball Hill). Further, the ice core record of Pinatubo $\Delta 33S$ gives a smaller signal (just above 0.5‰), granted it is still striking to preserve such S-MIF under an oxygenated atmosphere. We can imagine that under a much more weakly oxygenated atmosphere, such as ca. 2.3-2.2 Ga, there was an even greater possibility of generating a much larger S-MIF signal than that from Pinatubo. Again back to the Gallagher et al., 2017 idea: an alternative parsimonious explanation of the Carletonville MSI records is that a volcanogenic point source of sulfur might be able to make a significant impact on localized S-MIF records but not be of much consequence globally. Or even further, what if the brief returns of S-MIF younger than 2.3 Ga do not require anoxia?

L220-223: It follows that the periodic reappearances of S-MIF in the Timeball Hill formation may indicate a protracted, intermediate, and extremely sensitive atmospheric state that was uniquely susceptible to perturbation as oxygen contents vacillated around the threshold for S-MIF genesis, as opposed to a stepwise transition between bistable endmembers^{22,44-46}.

It could be good to emphasize that it is more than a question of "S-MIF genesis" by itself. Again, we could assume that some degree of S-MIF generation occurs throughout Earth history, even after the GOE (such as in the Pinatubo eruption example mentioned above). Considering this, such S-MIF only gets preserved under certain conditions. It could be analogous to how significantly positive O-MIF ($+\Delta 17O$) from atmospheric sulfates can contaminate carbonate associated sulfate records in carbonates that are subaerially exposed (e.g., Peng et al., 2014) but such significantly $+\Delta 17O$ does not make appreciable impact on global seawater sulfate.

Bryan Killingsworth
USGS

REFS:

Asael, D., Planavsky, N., Bellefroid, E., Hofmann, A. & Reinhard, C. in Goldschmidt (Paris, 2017).

Gallagher, M., Whitehouse, M.J. and Kamber, B.S., 2017. The Neoarchean surficial sulphur cycle: An alternative hypothesis based on analogies with 20th-century atmospheric lead. *Geobiology*, 15(3), pp.385-400.

Peng, Y., Bao, H., Pratt, L.M., Kaufman, A.J., Jiang, G., Boyd, D., Wang, Q., Zhou, C., Yuan, X., Xiao, S. and Loyd, S., 2014. Widespread contamination of carbonate-associated sulfate by present-day secondary atmospheric sulfate: evidence from triple oxygen isotopes. *Geology*, 42(9), pp.815-818.

Savarino, J., Romero, A., Cole-Dai, J., Bekki, S. and Thiemens, M.H., 2003. UV induced mass-independent sulfur isotope fractionation in stratospheric volcanic sulfate. *Geophysical Research Letters*, 30(21).

Reviewer #2 (Remarks to the Author):

This manuscript deals with the problem of mass-independent fractionation of sulfur in the sedimentary record, discussing the effect and timing of the crustal memory effect (CME) as compared to primary atmospheric signature. It uses a thorough compilation of previously published data as well as modelling to estimate the magnitude of this CME. It ends up comparing 2 sections in South Africa, with new measurements reported, and trying to explain the differences in sulfur isotope signature though their age are likely the same.

This is well written. The compilation will be helpful to many people.

The data are convincing and from my understanding the modelling approach is detailed and well explained. However I am a little confused by the different parts of the manuscript and the way they are linked.

1) The estimation of the cme with their updated compilation as well as the estimation of some biases in the sulfur isotope record is well explained.

2) The discussion of Turee Creek in the middle of the argument between Philippot on one side and Bekker on the other side, based on the validity of age constraints, leads to a less clear paragraph.

3) The discussion about the Timeball Hill Formation coming next added to my confusion and made the general message unclear.

The discussion about the bias in our archean sulfur measurements for the number of cratons or the age is interesting. I wonder how effective their correction is, because the scarcity of the Archean records makes it biased in itself. Is the corrected database closer to « reality » ? A discussion about the effect of the technique (bulk vs in situ, SF6 vs SIMS vs SO2) on the bias and on the subsequent cme would be interesting. Especially because the selection of pyrites with a reasonable size for in situ measurements induces a bias.

The estimation of the sea water sulfate reservoir and its isotopic signature shows spatial heterogeneity as pointed in the paragraph about Turee Creek. It varies a lot for Pilbara. What is the effect on other cratons (bigger or smaller cme ?).

In several places in the manuscript the authors claim that no S MIF in a sedimentary sample implies that there was no MIF generation in the atmosphere. $D_{33S} > 0.3$ requires an oxygen free atmosphere, but an oxygen free atmosphere does not imply that the sedimentary D_{33S} will be non zero.

So, line 214, « re-emergences of S-MIF » seems too affirmative. Seeing an anomaly proves that it s there, not seeing it does not mean it disappears.

The oxygen free feature of the atmosphere is a « global » signature, whereas the sedimentary record is the result of the interaction between the atmosphere and the local (oceanic or basinal) parameters.

Maybe the author could add a few words about the photochemical sulfide oxidation by iron as described in Hao et al 2022 (Sc Adv).

How could the authors adress the hypothesis that there was an anoxic atmosphere, with S-MIF generation, until 2.3Ga (or even 2.2Ga) and that it was locally recorded in sedimentary sulfides as sometimes $D_{33S} = 0$ and sometimes as $D_{33S} > 0.3$.

Reviewer #3 (Remarks to the Author):

Dear authors, dear editor,

Please find below my review for the manuscript submitted by Uveges and colleagues to Nature Communications.

Best regards

Guillaume Paris

In their manuscript entitled "Reconciling discrepant minor sulphur isotope records of the Great Oxidation Event", Uveges and colleagues re-explore the post GOE record of S-MIF.

They evaluate the reality, magnitude and reality of the crustal memory effect, by which S-MIF can be transferred from the crust to the ocean despite the presence of an oxygenated atmosphere. The main previous work on the topic (Reinhard et al) used a global average of the existing MIF dataset. Here the authors use an up-to-date dataset (10 years of additional data) and explore the influence of a weighted mean to calculate the CME. This is of importance as all data don't have the same geological meaning (1 spot of microanalysis vs. bulk analysis of a hand sample for instance) and in the literature, sometimes one pyrite grain can yield tens of datapoints. They start with the observation of a discrepancy between South Africa (no CME) and Australia (sustained CME). Thanks to different statistical approaches, they conclude that the Australian record can be explained by local CME due to more positive D33S values locally. Then, they explore the South African record and conclude that any post-GOE anoxic phase can only be very short lived in order to reconcile the existing data. If I read their paper correctly, they confirm the GOE age by Luo et al. but they don't clearly state it in the abstract or the paper.

The current draft is of interest to the readership of Nature Communication as it tackles a key question (timing of Earth's surface oxygenation) with a new approach and bring useful constraints to the community. It's a useful contribution. I have no doubt that this would become a highly cited paper on a strongly debated topic.

A few things should be adressed beofre publication.

The one missing element for a full review would be the synthetic database, so I'm trusting that it is correct and thorough. I am not able to evaluate the quality of the statistical work itself, so here as well I trust the authors and their work and hope that another reviewer will be more qualified than I am.

One element that I couldn't find in the current manuscript is the concentrations of O2 used by the authors to solve equation 3. How is it evaluated? How sensitive is the model to O2 levels? Could the authors provide the size of the sulfate reservoir calculated by their model and compare it to existing data? It seems that there is an underlying assumption of a homogeneous seawater D33S with a long lived residence time of sulfate in the ocean, couldn't it be otherwise? The d34S results could possibly be of interest as well, even though I know that mass dependent effects can make the task significantly more difficult.

Alternatively to the detrital pyrite scenario for the post-GOE data by Poulton et al.: could a positive spike in atmospheric oxygen concentration affect the weathering and deliver higher D33S to the ocean?

I also have one questions that is mostly curiosity. The authors establish that the putative return to an anoxic atmosphere are very brief and short, the only way to explain why Izon et al. couldn't find any. Pushing the reasoning further: what is the likelihood that those episodes are real? What is the likelihood that there could be more episodes that remain unsampled? Did Poulton et al. stroke gold and found all the episodes with the right sample at the right spot? Can this be evaluated?

In addition, what is the reason why the authors did not attempt to resample the same levels as Poulton et al.(or much closer to those samples)? Were the samples no longer available? As those are samples from a core, it should be possible to do so.

I found a few typos/details

Figure 2 : state the x axis is year of publication

Line 367, 382 : table SX1 instead of S1

Line 143 persistence of

In the SI documents : sometimes in the figure, samples are referred to as pre-2.3 ga or >2.3 Ga. It's easier for the reader if the authors remain consistent. Some words and references are highlighted or missing(eg. Line 177, 432, 525)

line 177 remove the question mark.

REVIEWER COMMENTS

Reviewer #1 (Remarks to the Author):

This study uses statistical tools and modeling to test the operation of a previously proposed global scale crustal memory effect (CME) of sedimentary recycling of sulfur isotope anomalies within the early Paleoproterozoic record of the Great Oxidation Event (GOE), ca. 2.5–2.3 billion years ago. It is a needed state of the art assessment of the efficacy of the mass-independent fraction sulfur isotope (S-MIF) record for tracking global signatures of the transition to an oxygenated atmosphere of greater than 10^{-5} PAL O_2 .

The results of the test of the CME are convincing, showing that the CME should not leave a globally resolvable sulfur isotope signature. While this study is worthy and compelling, there are some aspects that might deserve a further look, as suggested below.

We thank Dr. Killingsworth for his thoughtful review and overall enthusiasm toward our work. We have tried our best to address the issues that were raised, bolstering the discussion within the main text and appended supplement. Herein, we detail our response on a point-by-point basis, identifying where changes have been implemented within the manuscript and its ancillary files.

R1.1: Discrepancies between 2.45–2.2 Ga sulfur isotope records from the Turee Creek Basin (Pilbara craton) versus Transvaal Basin, and within sections from the Carletonville area in the Transvaal Basin (Kaalvaal craton) speak to the significant spatial sensitivity of the sulfur isotope record in the aftermath of the initial rise of oxygen ca. 2.45 Ga. After all, the records we are left are from variably restricted basins. We do not know what records might be considered representative of the global ocean versus more localized signals within restricted basins. A local CME, instead of only a global CME, may need to be tested to fully lay this issue to rest. Could the model of the CME be pushed further, to apply to the scale of a basin instead of the whole, global, seawater sulfate reservoir?

Yes, absolutely. To a first approximation, we can simulate the likely range of $\Delta^{33}S$ values communicated to an adjacent basin via a craton-specific weathering flux, thereby arriving at a rudimentary data-informed basin-specific $\Delta^{33}S$ threshold where atmospherically derived $\Delta^{33}S$ values can be distinguished from their crustally-inherited counterparts.

While we took this approach within our initial submission, simulating the Pilbara-derived weathered S-flux to make a direct comparison with the wealth of published data from the Turee Creek Basin (Fig. 4), Dr Killingsworth's review prompted us to extend our analysis to the two best represented cratons within the SSA database. Starting with the Kaapvaal Craton, after applying a LOESS fit to our bootstrap output, we simulate a CME with a maximum $\Delta^{33}S$ value of $0.84 \pm 0.6\text{‰}$ (1σ ; now shown in Fig S7), identifying $\Delta^{33}S$ values exceeding 1.5‰ as likely atmospheric derivatives. Contrastingly, the same approach applied to the Superior Craton returns a more muted CME ($\sim 0.15 \pm 0.2\text{‰}$, 1σ), fingerprinting values exceeding 0.5‰ as photochemical candidates. Consequently, these findings now offer numerical credence to the notion of a spatially variable CME and, as Dr Killingsworth alludes, and deemphasizes the importance CME as global construct. Interestingly, albeit at higher order, we hinted toward the possibility of spatial variability within the Turee Creek dataset (Philippot et al., 2018; Izon, unpublished data). Here, the deeper water facies (i.e., mud-and siltstones) within the Kazput Formation possess more muted $\Delta^{33}S$ values compared to their shallower equivalents (i.e., limestones); here, the former more closely resemble our best approximation of a global CME, while the latter presumably record a greater crustal influence. These observations, therefore, not only support inter-basinal heterogeneities but, in fact, imply intra-basinal variability will emerge as more detailed work is undertaken.

Importantly, however, we stress that our approach is imperfect: providing a *maximum* estimate assuming an isolated basin whose sulphate inventory is entirely derived from regional weathering. Abandoning this abstract stance to arrive at a more detailed basin/catchment specific analysis would require (i) a well constrained basin geometry, (ii) a knowledge of the basin's hydrographic communication with the global seawater inventory and (iii) an appreciation of relative importance of the weatherable terrestrial sulfur pool relative to the seawater sulphate inventory. Unfortunately, given that these parameters that are generally poorly known it is unlikely that more accurate basinal CMEs will be forthcoming. That said, given the larger proportion of elevated $\Delta^{33}S$

values within the database, the basins adjacent to the Pilbara and Kaapvaal cratons should bear witness to the most pronounced CME, rendering our estimates conservative and thus applicable to the wider record.

We have added components of this response to the main text at line 159-166.

R1.2: In another example of spatial variability, although the cited (ref. 38, lines 108–113) Torres et al., 2018 study is a clever one, S-MIF in contemporary sulfates in surface waters from Archean-aged terrains has been measured from S. Africa (Asael et al., 2017, Sulfur non mass dependent anomalies in modern river water of Archean catchment, Goldschmidt abstract). Both of those results are correct (though, taken individually, their interpretations may differ): weathering of Archean terrains today does and does not contribute S-MIF-bearing sulfate to rivers, likely depending on different catchment-specific factors. This could be even further evidence that the nature of sedimentary recycling of S-MIF (or, the crustal memory effect), if it shows up in the rock record, should be inherently localized.

This is an astute observation, and one we have now tried to expand upon and clarify within the main text. While, from R1.1, it follows that we absolutely expect the CME to be spatially variable, it is difficult to establish the spatial scale(s) on which such variability is expressed and how extreme the $\Delta^{33}\text{S}$ difference(s) might be. Although the routinely non-descript abstract language within Asael et al. (2017) precludes a detailed comment on their work, the apparently contrasting stance between the cited groups is presumably a function of integration. Interestingly, Torres et al. (2018) inadvertently highlight this with their dismissal of a 1.8‰ $\Delta^{33}\text{S}$ value found within a Kaapvaal-sourced groundwater sample (see their SI). Here, the authors presumed the sample's sulfur inventory averaged "a small volume of rock" and was, therefore, "not representative of the bulk Archean crust". While an extreme example, the same stance could be equally applied to analysis of fluvial sulfates. In essence, to communicate large $\Delta^{33}\text{S}$ anomalies to the weathered sulfate pool, the weathered sulfur must have an isotopically exotic source and, importantly, for this to be geologically important, these elevated $\Delta^{33}\text{S}$ values must escape the suffocating effect of dilution. Thus, as the catchment area becomes larger, one may hypothesize that the mixing of sulfur sourced from progressively more lithostratigraphic units should increasingly subdue the magnitude of the fluvial $\Delta^{33}\text{S}$ values, tending toward the catchment average downstream.

Our numerical approach offers two ways of exploring the importance of catchment integration. In essence, integration over progressively larger catchments is approximated via the isotopic sensitivity test presented in Fig. S6 (S3). Here, when small population sizes or, indeed, inadequate numbers of replicates are selected, the average $\Delta^{33}\text{S}$ value varies widely. As the sample size is increased, however, the mean $\Delta^{33}\text{S}$ value is seen to collapse toward its weighted equivalent. Taking a different stance, to ascertain the maximum range of $\Delta^{33}\text{S}$ values communicated to a

Fig. R1: (A) Sensitivity analysis/thought experiment utilizing only data from the Kaapvaal craton. Although an analogous time weighting to our isotope model was applied (Equ. 10), no temporal or spatial corrections were leveraged. This worst-case scenario is intentional, allowing principally the over-representation large $\Delta^{33}\text{S}$ values to be communicated to a localized weathered S-pool. (B) The standard deviations of means in (A).

hypothetical Kaapvaal hyper-localized sulfate pool, we performed an additional analysis whereby we applied a time-weighting to the Kaapvaal dataset without correcting for any spatial or temporal biases (Fig. R1). Here, even though these parameters are designed to exacerbate sampling biases and effectively over emphasize the late Neoproterozoic, as the population size increases, the grand mean trends to the weighted mean of the precursor time-weighted data. Importantly, for our purposes, even after the application of these extremely unrealistic parameters, we demonstrate there is effectively no chance of generating $\Delta^{33}\text{S}$ values that approximate those reported by Poulton et al. (2021; 2.9‰) from the Timeball Hill Formation (TBH Fm).

Finally, S-MIF can certainly be produced in an oxygenated atmosphere, with such records from as recent as 1991 from volcanic eruptions are found preserved in ice core (Savarino et al., 2003), provided such more recent volcanic sulfate does not differ in $\Delta^{36}\text{S}/\Delta^{33}\text{S}$ systematics versus Archean age sulfur. Nonetheless, it is their preservation that is critical to S-MIF records. As described, the key to S-MIF preservation is that the mixed phases of S-MIF bearing sulfur can be spatially separated and not homogenized at the Earth's surface. It could be that the analogy made between anthropogenic point-sources of atmospheric lead and the rain-out of this lead is indeed like the production and preservation of ancient S-MIF (Gallagher et al., 2017). Depending on the sensitivity of a particular Paleoproterozoic basin (due to sulfate concentration, sulfate weathering influx, facies controls, and biological controls), point sources of volcanogenic sulfate that do not have much long-lived global significance could make an outsized impact on S-MIF records from individual basins.

These are excellent points. To avoid repetition, we detail our response at the appropriate place below.

R1.3: General comment on the figures: the use of solid, filled, symbols make it difficult to appreciate the sample density on the plots. Could unfilled symbols be used instead?

Thank you. We have made the suggested change to Fig. 1 where inter-study comparison of sample densities was most useful/needed.

R1.4: L114–116: Inspection of the $\Delta^{33}\text{S}$ record shows that the last appreciably negative value occurs well before its positive $\Delta^{33}\text{S}$ counterpart (Fig. 1B). It would be nice to give the respective dates and S isotope values of the last appreciably negative and positive $\Delta^{33}\text{S}$.

This is a good suggestion and, indeed, something we contemplated prior to submission. Unfortunately, despite appearing straightforward, in practice this is not the case and we felt that assigning firm ages conveys an inappropriate level of certainty. In essence this is a deep time problem where the lack of precise age constraints and, often data-divorced, opinions propagate into 10s–100s of million-year uncertainties. Moreover, given the reports of elevated $\Delta^{33}\text{S}$ values within the TBH Fm. it is pertinent to distinguish between the youngest S-MIF derived from a definitively anoxic atmosphere, and the youngest positive S-MIF writ large. As we argue (line 222–225), it seems that the Rooihogte Formation represents a fundamental turning point in Earth's redox balance and, as such, we consider its large $\Delta^{33}\text{S}$ values to record the last vestige of a persistent Archean-like anoxic atmosphere. By contrast, the magnitude and structure of the $\Delta^{33}\text{S}$ values within the TBH Fm. appear to reflect atmospheric relapses acting against a more oxidized atmospheric state. Consequently, as it relates to our discussion of low-level oxidative weathering of pyrite under an S-MIF-yielding atmosphere, using the Rooihogte Formation as the youngest positive S-MIF is more apt, but is not correct in the strictest sense.

In our contribution we generally adopt the nearest adjacent age constraints to assign maximum and minimum depositional age constraints (Fig. 1; Table 1). While this approach is practical, it results in a large uncertainty that is difficult to quantify. Applying this approach to place maximum depositional constraint on the Rooihogte Formation, for example, leverages a tuff-derived U–Pb zircon age ($2480 \pm 6\text{Ma}$; Nelson et al., 1999) from the unconformably underlying Penge Iron Formation. Given that this is widely different from the precise Re–Os isochron age derived from the conformably overlying TBH Fm., we use the sequence boundary that bisects the Transvaal succession for our adopted maximum age of the Rooihogte Formation. For most of the record, however, the paucity of well-defined sedimentary relationships prevents this approach, leaving huge uncertainties that allow the coexistence of sometimes vastly conflicting models.

Returning to Dr Killingsworth's specific point, the youngest $\Delta^{33}\text{S}$ value statistically lower than -0.3‰ is found within the Boolgeeda Iron Formation, Turee Creek Group, Australia. As discussed in the supplement (§S1.1), however, the upper bounding ages of the Turee Creek Group are contested. If we adhere to the age model forwarded by Philippot et al. (2018), taking the mean of the adjacent age constraints (2450 and 2340 Ma) gives a mean age of 2395 Ma. Uncertainties surrounding the minimum age constraint aside, the true age of the Boolgeeda Iron Formation that hosts the negative $\Delta^{33}\text{S}$ values is likely much closer to 2450 Ma given that the upper age constraint is separated by 1.5 km of stratigraphy. A 2450–2420 Ma age for these negative $\Delta^{33}\text{S}$ values brings them in line with the Heinskop/Rooinnekke Formations, Ghaap Group, South Africa. Although these age constraints are also contested, these units contain some of the most pronounced negative $\Delta^{33}\text{S}$ values within in the record, dating to between 2436–2426 Ma (Fig. S2; Warke et al., 2020b). The youngest positive $\Delta^{33}\text{S}$ values prior to the TBH Fm. are found within the Rooihogte Formation (up to $+8.76\text{‰}$). The best available age constraints on the Rooihogte formation yield a maximum depositional age of 2353 ± 18 Ma and a minimum age of 2316 ± 7 Ma, resulting in a mean age of 2334 Ma (Fig. 1), aligning with the sedimentation-rate-derived age (2330 Ma) adopted by Luo et al. (2016). So, in summary, depending on the age model, there is 60–100-million-year spread between the youngest large negative and positive $\Delta^{33}\text{S}$ values. While we lean towards the larger of those two estimates, a case has been made that the separation was both much smaller (Warke et al., 2020a) and much larger (>200 Myr, Poulton et al., 2021).

A version of this discussion has been added to the supplement (§S1.2.3) and the 60 Myr age gap to main text line 108–111).

R1.5: L171-176: Consequently, provisionally accepting previous interpretations that the elevated $\Delta^{33}\text{S}$ values seen within the Timeball Hill Formation (Fig. 1C) do in fact faithfully record atmospheric chemistry²⁵, we ask the question how can we reconcile the apparently discrepant Carletonville MSI records? The most parsimonious explanation is that these intervals must be stratigraphically isolated and therefore represent geologically brief returns to an oxygen-free atmospheric state capable of S-MIF genesis and export²⁵.

If the occurrence of S-MIF in Timeball Hill is indeed a contemporaneous atmospheric signal, does this really require an oxygen-free atmosphere? S-MIF can certainly be produced in an oxygenated atmosphere; however, its preservation is another challenge. For example, the 1991 Pinatubo volcanic eruption resulted in sulfur with elevated $\Delta^{33}\text{S}$ preserved in ice core (Savarino et al., 2003). The caveat is that the $\Delta^{36}\text{S}/\Delta^{33}\text{S}$ isotope systematics are different in the sulphate ice core record of the Pinatubo eruption compared to Archean $\Delta^{36}\text{S}/\Delta^{33}\text{S}$ signatures, whereas the former gave a steeper slope (though, interestingly, at -4.3 this MIF slope is close to the "MDF" slope of -3.2 identified by Poulton et al., 2021 in Rooihogte–Timeball Hill). Further, the ice core record of Pinatubo $\Delta^{33}\text{S}$ gives a smaller signal (just above 0.5‰), granted it is still striking to preserve such S-MIF under an oxygenated atmosphere. We can imagine that under a much weaker oxygenated atmosphere, such as ca. 2.3–2.2 Ga, there was an even greater possibility of generating a much larger S-MIF signal than that from Pinatubo. Again, back to the Gallagher et al., 2017 idea: an alternative parsimonious explanation of the Carletonville MSI records is that a volcanogenic point source of sulfur might be able to make a significant impact on localized S-MIF records but not be of much consequence globally. Or even further, what if the brief returns of S-MIF younger than 2.3 Ga do not require anoxia?

Thank you for reminding us of the model advocated by Gallagher et al. (2017). Upon reflection, however, there are several issues with this model that prevent us from favoring its extension to our data:

The model advanced Gallagher et al (2017) inflates $\Delta^{33}\text{S}$ values via two-step volcanically induced photochemical processing, whereby previously photolyzed sulfur is re-injected into the atmosphere by volcanoes via sedimentary incorporation into the mantle. As their model was advocated as an alternate take on the Neoproterozoic sulfur cycle, the preservation of the S-MIF signal within the record is reliant the coexistence of multiple atmospheric exit channels that separate the photochemical S-pools in space. A volcanic eruption against an oxygenated atmospheric backdrop, however, invalidates this prerequisite because of the presumed oxidation of the S^0 phase. If true, perhaps modern volcanogenic $\Delta^{33}\text{S}$ signatures can be used as an analogue to understand how such an event might play out in the rock record. Ice core sulfates feature volcanogenic $\Delta^{33}\text{S}$ anomalies not only because they escape dilution with terrestrial sulfate, but because of differential rainout that remains intact

within pristine ice cores. Here, rather than spatially separating the isotopically distinct S-phases, the S-MIF bearing sulfates are essentially separated in time by the relatively rapid accumulation of the snow/ice pack. Mechanistically, because atmospheric H_2SO_4 production proceeds more rapidly via a SO_3 , rather than a SO_2 , mass-balance predicts that atmospheric sulfates will transition from positive to negative $\Delta^{33}\text{S}$ values as time progresses (e.g., Baroni et al., 2007). Indeed, with the advent of MC-ICPMS, this characteristic positive-to-negative $\Delta^{33}\text{S}$ evolution has now been demonstrated to occur over the passage of several years (e.g., Burke et al., 2019).

Consequently, we arrive at two principle objections to a large volcanic point-source of sulfur as the source of the observed signals under a continuously oxidizing backdrop:

- 1) The TBH $\Delta^{33}\text{S}$ values are preserved in marine sedimentary rocks and, therefore, must have encountered a dissolved sulphate phase (though its true it could be spatially variable and low), rather than be directly entombed within ice.
- 2) It is effectively impossible that we would be able to resolve such high-order signals given the processes involved in partitioning sulfur within the early diagenetic environment and the degree of compaction of these sediments have experienced.

For the volcanic point-source model to work under an oxidized backdrop, the starting $\Delta^{33}\text{S}$ composition of the emitted aerosols would need to be large. Using our database we can explore the feasibility this idea within the wider Neoproterozoic record (*sensu* Gallagher et al., 2017) as well as its application to TBH Fm. While Gallagher and colleagues envisage Neoproterozoic volcanic sulfur carrying strongly positive $\Delta^{33}\text{S}$ values (i.e., 3.5–9.46‰), interrogation of our database is hard to reconcile with ^{33}S -enrichments of this magnitude (Table R1–R3). The average composition of the pre-Neoproterozoic record is 0.15‰, increasing slightly to and 0.19‰ when the sample averaged database is used (Table R1). Thus, even the suggested ^{33}S -depleted sulfate phase (3.5‰) is outside of the 97.5th quantile (q975, Table R1), meaning that more than 97.5% of the *non-bias corrected* pre-Neoproterozoic record carries more subdued $\Delta^{33}\text{S}$ values than that envisaged for their negatively fractionated endmember. Given that Wilson-cycle-timescales would implicate Mesoproterozoic-aged sulfur as the predominant source to Neoproterozoic aged subduction zones, the clearly muted $\Delta^{33}\text{S}$ values from this Era compared to the rest of the Archean–Paleoproterozoic record (Fig. 1., Table R2, Fakrae et al., 2018) further confound this issue. Further, although we have reduced the positive $\Delta^{33}\text{S}$ skew in the wider S-isotope record, its continued presence requires an accompanying negative $\Delta^{33}\text{S}$ sink. Conventional understanding of the S-MIF record states that sulfate is the negative endmember (Farquhar et al., 2013; Farquhar and Wing, 2003; Jamieson et al., 2006; Ono et al., 2003; Reinhard et al., 2013), and thus its influence should be the strongest in deep-sea sediments and oceanic crust that would be the presumed source of sulfur to arc magnetism. Consequently, these observations conflict with the Gallagher model by requiring not only that recycled S-MIF-bearing sulfur was the dominant sulfur source in the erupted aerosols, but also that its $\Delta^{33}\text{S}$ composition be sourced from the extreme tail of the known record, in a reservoir that is not even expected to host a significant amount of that tail. Even the TBH data (Poulton et al., 2021), showing a maximum $\Delta^{33}\text{S}$ of 2.9‰, which would be at least partially diluted by the existing SWSR, still fall victim to the same issues.

A potentially more feasible corollary could be drawn from the recent recognition of small $\Delta^{33}\text{S}$ values observed in sediments immediately after the K-Pg impact event (Junium et al., 2022), which has the potential serve as a much larger instantaneous point-source of sulfur. The fact that these values can be resolved points toward the potential for some form of preservation in the rock record, however, the $\Delta^{33}\text{S}$ values are universally negative and small. Ignoring these issues and the repeated nature of the $\Delta^{33}\text{S}$ excursions in the TBH Fm., it is unreasonable to suggest that such a cataclysmic event could have occurred without leaving impact-related sedimentary indicators (e.g., spherules, tsunamites/tempestites, iridium anomaly etc.).

There is also the issue of slopes; both the $\Delta^{36}\text{S}/\Delta^{33}\text{S}$ and $\delta^{34}\text{S}/\Delta^{33}\text{S}$ of ice core sulfates and, indeed, the post K–Pg impact data are distinctly different from the Archean Reference Array. Further, the S-MIF in the TBH Fm. do not align with the "volcanic array" described by (Philippot et al., 2012). The fact that the TBH S-MIF intervals align with the Archean array (though the source of that slope is still not fully understood) detracts from the various volcanic/point source templates that we can compare it with. Thinking more widely, perhaps large-scale LIP tye volcanism could have released sufficient reducing power to consume O_2 temporally reinstating Archean-like

photochemistry (Wogan et al., 2022). Or, given a substantially large volcanic eruption, the point source model may again be feasible, with the caveat that O₂ was drawn down enough to allow for separation of two distinct phases. Overall, both scenarios are compatible with our proposed generalized model of an intermediate/low oxygen concentration prone to periodic returns to an anoxic atmosphere.

We have added a shortened version of this response to the main text (line 226–247) and in a new section (S3) of the supplementary discussion. Tables R1–R3 are added to the supplement as Tables S4.1–4.3.

R1.6:L220–223: It follows that the periodic reappearances of S-MIF in the Timeball Hill formation may indicate a protracted, intermediate, and extremely sensitive atmospheric state that was uniquely susceptible to perturbation as oxygen contents vacillated around the threshold for S-MIF genesis, as opposed to a stepwise transition between bistable endmembers^{22,44–46}. It could be good to emphasize that it is more than a question of “S-MIF genesis” by itself. Again, we could assume that some degree of S-MIF generation occurs throughout Earth history, even after the GOE (such as in the Pinatubo eruption example mentioned above). Considering this, such S-MIF only gets preserved under certain conditions. It could be analogous to how significantly positive O-MIF (+Δ¹⁷O) from atmospheric sulfates can contaminate carbonate associated sulphate records in carbonates that are subaerially exposed (e.g., Peng et al., 2014) but such significantly +Δ¹⁷O does not make appreciable impact on global seawater sulphate.

This is a good point, particularly given the preceding discussion of volcanic point sources. We have added a qualifier of genesis/preservation to the text (line 255).

Reviewer #2 (Remarks to the Author):

This manuscript deals with the problem of mass-independent fractionation of sulfur in the sedimentary record, discussing the effect and timing of the crustal memory effect (CME) as compared to primary atmospheric signature. It uses a thorough compilation of previously published data as well as modelling to estimate the magnitude of this CME. It ends up comparing 2 sections in South Africa, with new measurements reported, and trying to explain the differences in sulfur isotope signature though their age is likely the same. This is well written. The compilation will be helpful to many people. The data are convincing and from my understanding the modelling approach is detailed and well explained. However, I am a little confused by the different parts of the manuscript and the way they are linked.

- The estimation of the cme with their updated compilation as well as the estimation of some biases in the sulfur isotope record is well explained.
- 2) The discussion of Turee Creek in the middle of the argument between Philippot on one side and Bekker on the other side, based on the validity of age constraints, leads to a less clear paragraph.
- 3) The discussion about the Timeball Hill Formation coming next added to my confusion and made the general message unclear.

Our contribution seeks to reconcile several inconsistencies within the S-MIF record that combine to hinder our understanding of the GOE. Given the ground we needed to cover, compounded by the journal’s space constraints, our message must have been obscured. We apologize for the ambiguity and thank the reviewer for their patience. In the revised manuscript we have attempted to clarify the narrative, relocating the discussion surrounding the Bekker/Poulton age constraints within the SI (S1.1) where more detail can be provided. More generally we have revisited the manuscript’s structure to smooth our thought process and increase readability throughout.

R2.1: The discussion about the bias in our Archean sulfur measurements for the number of cratons or the age is interesting. I wonder how effective their correction is because the scarcity of the Archean records makes it biased. Is the corrected database closer to « reality » ?

It is true that the paucity of Archean-aged rocks reflects a bias in our understanding that may, or may not, be reasonably compensated for. While we do not really see a mechanism where the present-day inventory of Archean rocks has resulted from a selective preservation, how the ravages of time have influenced the sulfur isotope record really requires a more complete mechanistic understanding of S-MIF production and preservation. That said, after recalling the overwhelming overrepresentation of a handful of formations, we

contend that our correction scheme *is*, in fact, closer to reality when compared with existing estimates that leverage a clearly inadequate cumulative mean.

R2.2: A discussion about the effect of the technique (bulk vs in situ, SF₆ vs SIMS vs SO₂) on the bias and on the subsequent CME would be interesting. Especially because the selection of pyrites with a reasonable size for in situ measurements induces a bias.

A valid point and, indeed, one we endeavored to account for in our approach via the “Spot Sample Average” database. While this is fully described in the main text/methods we reiterate that all intra-sample spot measurements (e.g., SIMS) were averaged, reducing each sample to a single number that effectively corrects for sample density biases. Nevertheless, while this removes the over representation of grain-scale analysis, we concede that it does not account for systematic textural biases arising through the initial motivation of the study and/or the amenability of specific pyrite grains to spatial analysis. This is, in fact, empirically demonstrated by studies that show that SIMS $\Delta^{33}\text{S}$ averages do not always mirror their bulk counterparts (Farquhar et al., 2013; Izon et al., 2022; Johnson et al., 2013; Muller et al., 2017; Ono et al., 2009; Philippot et al., 2018). In detail, however, the magnitude and sign of the offset between bulk- and SIMS-derived data makes it difficult to assess how such a bias impacts our analyses. Moreover, these clearly sample-specific differences preclude an *en masse* correction that, in fact, likely would induce more uncertainty than it would solve.

Interrogation of our pre-2.3 Ga database allows us to plot $\Delta^{33}\text{S}$ frequency distributions by analysis type (Fig. R2). Here, while a cursory inspection of Fig. R2 appears to disclose analytical biases, closer inspection alongside Table R4, reveals that these analytical differences appear to be an artifact of a small population size and/or the application to a specific geological succession/phase rather than the technique itself. Perhaps most conspicuously, the $\Delta^{33}\text{S}$ distribution derived from multi-collector inductively coupled mass spectrometry (MC-ICPMS) lacks a pronounced mean around zero and features a tail extending to extremely positive $\Delta^{33}\text{S}$ values that is unparalleled elsewhere. In detail, however, these MC-ICPMS data are exclusively analyses of carbonate associated sulfate (CAS) extracted principally from the Gamohaan and Reveilo formations (71/94 samples). When comparing analytical approaches where more than 100 samples have been analyzed (e.g., Bulk EA-CF-IRMS, Bulk SF₆, SHRIMP SI and SIMS), however, very little difference in their $\Delta^{33}\text{S}$ distributions can be seen (Fig R2, Table R4). Moreover, this similarity holds between spatially resolved and bulk approaches, irrespective of whether spot averaging has been performed. On closer inspection, there are some slight differences between the means from the bulk- and spot-derived datasets, however, these are largely the result of temporal and spatial biases that we describe and correct for within our model. So, at least at the broadest scale, the actual analytical approach is less important than the overrepresentation of specific formations. An intra-laboratory method comparison and validation study, similar to that of Ushikubo et al., (2014), using a set of common samples analyzed by a range of preparation and analytical techniques would be better able to estimate any methodological bias.

R2.3: The estimation of the sea water sulfate reservoir and its isotopic signature shows spatial heterogeneity as pointed in the paragraph about Turee Creek. It varies a lot for Pilbara. What is the effect on other cratons (bigger or smaller cme ?).

This is a great question that we have tried to better work into the main text of the paper (line 159–166) and the SI. For a full answer please refer to R1.1 above

R2.3: In several places in the manuscript the authors claim that no S MIF in a sedimentary sample implies that there was no MIF generation in the atmosphere. $D_{33S} > 0.3$ requires an oxygen free atmosphere, but an oxygen free atmosphere does not imply that the sedimentary D_{33S} will be non-zero. So, line 214, « re-emergences of S-MIF » seems too affirmative. Seeing an anomaly proves that it's there, not seeing it does not mean it disappears. The oxygen free feature of the atmosphere is a « global » signature, whereas the sedimentary record is the result of the interaction between the atmosphere and the local (oceanic or basinal) parameters.

In essence this is true and aligns well with the fact that a large percentage of the S-MIF record is close to zero (§S2.1, Fig. S3). Compilation and interrogation of a database, however, offers statistical power to test the reviewers counter claims. Here, leveraging our database, if we exclude the data from the Turee Creek Group, which we argue are the result of a localized CME in an isolated basin, the overwhelming majority of the 2.3–2.1 Ga Δ^{33S} record (> 90%), spanning multiple cratons, displays Δ^{33S} values between -0.3 and 0.3 (Fig. R3). This is in stark contrast with the pre-2.3 Ga record where only ~40% of the record is between -0.3 and 0.3 and more closely mirrors the post-2.1 Ga dataset (Fig. R3). Moreover, when comparing the Timeball Hill Formation with units deposited prior to 2.2 Ga that feature greater than 20 QSI analyses, the only units that approximate the same median and interquartile range (IQR) are the Duitsland, Polisarka, Gordon Lake and, to a lesser extent, the Pecors formations (Fig. R4). Aside from the Pecors Formation, these lithostratigraphic units are either younger than ~2.33 Ga and/or have been associated with a rise in oxygen (Fig. R4). By contrast, the vast majority of applicable units either have a median sufficiently outside of 0 or have large enough interquartile ranges to indicate that there is a significant presence of larger Δ^{33S} values from which to infer an anoxic atmosphere (Fig. R4). This combined with the fact that there is no broad correlation of Δ^{33S} values with facies, suggests the elevated Δ^{33S} values seen within the Timeball Hill Formation record rare returns to S-MIF production superimposed on an oxic backdrop, rather than the Timeball Hill Formation featuring minor S-isotope systematics that contrast with the rest of the record. So, in summary, while the reviewer is correct when they state that a value of less than |0.3‰| in isolation does not indicate an oxic atmosphere, the overwhelming preponderance of these MDF-like Δ^{33S} values do. Arguing otherwise would require a significant departure from our understanding of S-MIF

Fig. R3: Distribution of Δ^{33S} data, ascending from the bottom panel shows data older than 2.3 Ga, those sourced from between 2.3 and 2.1 Ga (excluding the Turee Creek Group) and those younger than 2.1 Ga.

Fig. R4: Y-axis is the formation-specific median Δ^{33S} value, while vertical bars represent the IQR between the 25th (lowerbound) and the 75th (upperbound) percentiles. The horizontal line is 0‰. The Timeball Hill Formation is in red.

production/preservation, contradict interpretations of countless independent geochemical datasets and require a rewriting of our understanding Earth history.

A shortened version of this response is included in the Manuscript (line 213–225) and Fig. R3 is included in the SI as Fig. S8.

R2.4: Maybe the author could add a few words about the photochemical sulfide oxidation by iron as described in Hao et al 2022 (Sc Adv).

The mechanism proposed by Hao et al. (2022) is an interesting one and could have dramatic implications for our understanding of Archean trace element budgets and the size of the SWSR inventory in deep-time. Though, as it would be drawing from the same pool of weatherable sulfur it should not alter our maximum estimate of a CME syn/post-GOE. It may, however, promote more positive $\Delta^{33}\text{S}$ values prior to the rise of oxygen (e.g., bringing the pre-2500 Ma estimate shown in Fig. 3 closer to or above zero). If photochemical oxidation was a volumetrically significant source of sulfate relative to early oxidative pyrite weathering it may affect our estimate of the SWSR $\Delta^{33}\text{S}$ and/or the magnitude of the expressed global CME, by moving it closer to the weathered estimate (i.e., the red curve would more closely approximate the blue curve in Fig. 3 but this would be a relatively minor shift. A brief version of this response has been added to our methods summary (line 341–345).

R2.5: How could the authors address the hypothesis that there was an anoxic atmosphere, with S-MIF generation, until 2.3Ga (or even 2.2Ga) and that it was locally recorded in sedimentary sulfides as sometimes $D^{33}\text{S}=0$ and sometimes as $D^{33}\text{S} > 0.3$.

This is addressed above. To reiterate, however, outside of the Turee Creek Group, which we argue is influenced by a regional and pervasive CME (or is potentially significantly older, §S1) the 2.3–2.1 Ga $\Delta^{33}\text{S}$ record appears entirely different to its precursor (Fig. R3). This fundamental overhaul, manifest within a large data compilation, speaks to shift in S-MIF export and, most parsimoniously, reflects atmospheric oxygenation.

Reviewer #3 (Remarks to the Author):

In their manuscript entitled "Reconciling discrepant minor sulphur isotope records of the Great Oxidation Event", Uveges and colleagues re-explore the post GOE record of S-MIF. They evaluate the reality, magnitude, and reality of the crustal memory effect, by which S-MIF can be transferred from the crust to the ocean despite the presence of an oxygenated atmosphere. The main previous work on the topic (Reinhard et al) used a global average of the existing MIF dataset. Here the authors use an up-to-date dataset (10 years of additional data) and explore the influence of a weighted mean to calculate the CME. This is of importance as all data don't have the same geological meaning (1 spot of microanalysis vs. bulk analysis of a hand sample for instance) and in the literature, sometimes one pyrite grain can yield tens of datapoints. They start with the observation of a discrepancy between South Africa (no CME) and Australia (sustained CME). Thanks to different statistical approaches, they conclude that the Australian record can be explained by local CME due to more positive $D^{33}\text{S}$ values locally. Then, they explore the South African record and conclude that any post-GOE anoxic phase can only be very short lived to reconcile the existing data. If I read their paper correctly, they confirm the GOE age by Luo et al. but they don't clearly state it in the abstract or the paper.

The current draft is of interest to the readership of Nature Communication as it tackles a key question (timing of Earth's surface oxygenation) with a new approach and bring useful constraints to the community. It's a useful contribution. I have no doubt that this would become a highly cited paper on a strongly debated topic.

We thank Dr Paris for his appraisal and generally positive reflection of our work. His summary is a good precis of our manuscript and our overall stance. The only clarification we would like to make is that our analyses do not necessarily support, nor refute the age of the GOE advocated by Luo et al. (2016) *sensu* Luo et al. (2016). As discussed in the supplement (§S1.2), there is scope, albeit weakly supported, for there to be an earlier oxidation "event/episode" at ca. 2.45 Ga. We do think, however, that our analyses implicate a tipping point within the Earth system, whereby the background atmospheric state had evolved sufficiently to restrict S-MIF export prior to deposition of the Timeball Hill Formation. Against this backdrop, atmospheric dynamics would operate in an opposite sense relative to the background state.

R3.1: The one missing element for a full review would be the synthetic database, so I'm trusting that it is correct and thorough.

We apologize. The full database, as well as iterations of the TA and CA synthetic database used in the modelling analyses, are now provided in a zip file containing a summation of the code and data used in this study (New_MIF_database.xlsx, SSA_database.csv, MIF_boot_sample.csv, and Copy_of_MIF_boot_sample_cratons.csv). The full database, and a user-friendly app, will be included as a supplementary file if the manuscript is deemed publication worthy.

R3.2: I am not able to evaluate the quality of the statistical work itself, so here as well I trust the authors and their work and hope that another reviewer will be more qualified than I am. One element that I couldn't find in the current manuscript is the concentrations of O_2 used by the authors to solve equation 3. How is it evaluated? How sensitive is the model to O_2 levels?

The oxygen concentration used to calculate Equ. 3 and, indeed, scale the bootstrap sampling parameters within the isotope model was derived from the outputs of the geochemical model described by Luo et al. (2016). This is clarified on line 307–312. In essence the model tracks the relevant fluxes of carbon, hydrogen, oxygen, sulfur, and methane in the prelude to the GOE. Combining these O_2 and S fluxes with our new isotope model, we then temporally solve Equ. 3–9. Within the bootstrap subroutine of the isotope model, the impact of oxidative weathering is effectively expressed as a probability weighting that progressively increases the likelihood of pyrite contributing to the weathered sulfur flux as oxygen concentrations increase. After a certain threshold, however, pyrite oxidation dominates and the $\Delta^{33}S$ composition of the weathered S flux becomes a function of the bias-corrected dataset that the model is prescribed.

Thinking about this in real-world-space, while the switch from a sulfate- to sulfide-dominated weathering régime is sensitive to pO_2 , in reality the maximum expression of the CME is not particularly oxygen sensitive unless it is continually maintained at the threshold for oxidative pyrite weathering after the geological loss of S-MIF. Such a scenario, however, is not easily produced using the model advocated by Luo et al. (2016). Importantly, given that the pO_2 threshold for S-MIF export is higher than that needed to induce pyrite weathering (Johnson et al., 2021), such a scenario is not realistic. Consequently, once pO_2 has risen sufficiently to curtail S-MIF export, the pO_2 is high enough to effectively ignore the oxygen sensitivity of the model.

R3.3: Could the authors provide the size of the sulfate reservoir calculated by their model and compare it to existing data? It seems that there is an underlying assumption of a homogeneous seawater $D^{33}S$ with a long-lived residence time of sulfate in the ocean, couldn't it be otherwise? The $d^{34}S$ results could possibly be of interest as well, even though I know that mass dependent effects can make the task significantly more difficult.

The modelled $[SO_4^{2-}]$ output transitions from $\sim 10\mu M$ to in-excess-of 1mM over the course of the GOE transition (see Luo et al., 2016), which is consistent with concentrations proposed by others (e.g., Canfield and Farquhar, 2009; Planavsky et al., 2012; Williford et al., 2011). Such low sulphate levels go together with a heterogeneous seawater sulphate reservoir and is now clarified alongside our discussion of a regionally variable CME (line 136–139 and 226–227).

In terms of $\delta^{34}S$, the modelled output is roughly 2‰ surrounding the GOE. Recalling that the Rooihoogte–Deutschland succession possesses a $\delta^{34}S$ range exceeding 55‰ (Izon et al., 2022; Luo et al., 2016; Poulton et al., 2021) and, even at the grain-scale, a single sample can feature $\delta^{34}S$ values spanning more than 20‰ (Izon et al., 2022), such a near-zero $\delta^{34}S$ value is of little concern or utility.

R3.4: Alternatively, to the detrital pyrite scenario for the post-GOE data by Poulton et al.: could a positive spike in atmospheric oxygen concentration affect the weathering and deliver higher $D^{33}S$ to the ocean?

This is an interesting suggestion, however, the idea that the circum-zero $\Delta^{33}S$ values record an anoxic (or low oxygen) atmospheric state and the $\Delta^{33}S$ spikes signified more a more oxygen-replete atmospheric state would be a significant departure from our understanding of S-MIF systematics. We feel such a reversal is unlikely for several reasons: First, the magnitude of some of Poulton and colleagues' $\Delta^{33}S$ data (upwards of 2.9‰) are well outside of what we calculate is feasible for a basin specific CME, which is likely much, much, less than 1.5‰ (See R1.1 and new Fig. S7). Second, the nature of the data from the Turee Creek Group is more consistent with a long-lived smooth evolution of a regional CME, rather than periodic spikes as seen in the Poulton record (Fig. 1c). Lastly, as discussed in response to a comment above (See R2.3), the overall dominance of $\sim 0\%$ values in the TBH Formation is a distinct shift from the pre-2.3 Ga record. Assimilating this tripartite argument, it seems much more likely that the TBH-housed $\Delta^{33}S$ spikes represent anoxic events superimposed on a mildly oxic backdrop rather than the inverse.

R3.5: I also have one questions that is mostly curiosity. The authors establish that the putative return to an anoxic atmosphere are very brief and short, the only way to explain why Izon et al. couldn't find any. Pushing the reasoning

further: what is the likelihood that those episodes are real? What is the likelihood that there could be more episodes that remain unsampled? Did Poulton et al. stroke gold and found all the episodes with the right sample at the right spot? Can this be evaluated?

This is an excellent question and one that we have struggled with. First, credit where it is due; Poulton et al. confirmed their TBH non-zero $\Delta^{33}\text{S}$ numbers by two different methods (SO_2 and SF_6), conducting their analysis in three different labs. Consequently, we have no grounds to fault their data. Further, should they send us their Ag_2S we'd fully expect to find the same elevated $\Delta^{33}\text{S}$ values within our respective uncertainties. Consequently, the only way that the TBH non-zero $\Delta^{33}\text{S}$ values could conceivably be a laboratory-based artifact would be to invoke cross-contamination with S-MIF-bearing Rooihogte samples and, even then, back-of-envelope mass-balance calculations essentially render that an impossibility.

Inverting the question, however, is a good one and one we've widely encountered from the skeptical community. In essence, inverting the results of our synthetic sampling experiment offers some insight (Fig. 5a). At very small S-MIF windows the likelihood of reproducing their hit rate (~20%) is exceedingly low. As the S-MIF window size increases the likelihood increases, with the most likely window size being somewhere between 2 and 5 meters (Fig. 5a). Our Bayesian analysis provides the most likely estimate of episode size in core (1.42 m), which is replicated in our bootstrap sampling experiment where an equal likelihood is returned in a 60-sample population. That said, neither event was very likely, with both scenarios existing in the extreme tails of the resultant bootstrap distribution (Fig. 5a). Given the inferred brevity of the S-MIF intervals, we would certainly expect that, if they are real, that there would be more of them distributed throughout the succession. Exploring this hypothesis, however, requires an even higher resolution study than that performed by Izon and Luo et al. (2022).

It is important to note that all the simulated scenarios assume quasi-random sampling practices. If the samples taken by Poulton et al. invalidate this assumption, perhaps by selective sampling of a given texture (?), then this might explain their extraordinary hit rate.

R3.6: In addition, what is the reason why the authors did not attempt to resample the same levels as Poulton et al. (or much closer to those samples)? Were the samples no longer available? As those are samples from a core, it should be possible to do so.

This is a great question and something we have tried to secure funding to do. Unfortunately, our efforts have proven unsuccessful. While seemingly straightforward, accurately resampling core is inherently difficult. Since recovery, popular cores are sampled by multiple investigators with varying degrees of precision. While the depths of their samples are typically recorded within notebooks or sample bags, the core repository is rarely privy to this information making it difficult to ascertain what was taken from where and by whom. Perhaps more frustratingly, as time progresses, and cores move within their boxes, workers concentrate their sampling efforts around the drillers' annotations making it increasingly more difficult to integrate sample sets. To conduct a meaningful comparison study would require a thorough logging and resampling of the EBA cores. Besides the obvious financial implications, a combination of COVID and visa restrictions has prevented us from resampling the EBA cores in person, leaving us with the residual material that was collected as part of a previous sampling campaign. Luckily, some of this material was very close to several of Poulton and colleagues' S-MIF-bearing Timeball Hill Formation samples and thus could be leveraged to fortify our numerically grounded arguments.

R3.7: Figure 2: State the x axis is year of publication. Fixed. Thank you

R3.8: Line 367, 382: Table SX1 instead of S1. Thank you for drawing our attention to this ambiguity. In this case, SX1 and SX2 are intended to refer to the different sheets, labeled SX1 and SX2, in the supplementary excel file, rather than tables within the Supplementary Information (.dox). We have added the term Supplementary Data File, where necessary, to clarify.

R3.9: Line 143 persistence of Fixed. Thank you.

R3.10: In the SI documents : sometimes in the figure, samples are referred to as pre-2.3 ga or >2.3 Ga. It's easier for the reader if the authors remain consistent. Thank you. We have now fixed the errant figure (Fig. S4)

R3.11: Some words and references are highlighted or missing (e.g., lines 177, 432, 525) Fixed.

R3.12: line 177 remove the question mark. Fixed.

Tables

Mean	SD	Median	q2.5	q25	q75	q95	q97.5
pre-Neoarchean							
0.148	1.09	-0.06	-1.16	-0.42	0.4	2.07	2.85
SSA pre-Neoarchean							
0.185	0.89	0.023	-1.11	-0.27	0.45	2.05	2.64

Table R1: Descriptive statistics of $\Delta^{33}\text{S}$ database utilising only pre-Neoarchean aged samples. SD = standard deviation, q# = the #th quantile.

Mean	SD	Median	q2.5	q25	q75	q95	q97.5
Mesoarchean							
-0.116	0.68	-0.08	-1.77	-0.366	0.1	0.9	1.26
SSA Mesoarchean							
0.0581	0.647	0	-1.07	-0.2	0.2	1.2	1.3

Table R2: Descriptive statistics of $\Delta^{33}\text{S}$ database utilising exclusively Mesoarchean aged samples. SD = standard deviation, q# = the #th quantile.

Mean	SD	Median	q2.5	q25	q75	q85	q95	q97.5
pre-2.3 Ga								
0.739	1.97	0.151	-1.58	-0.21	0.93	1.95	5.03	6.77
SSA pre-2.3 Ga								
1.11	2.06	0.3	-1.17	-0.044	1.72	3.05	5.82	6.79

Table R3: Descriptive statistics of $\Delta^{33}\text{S}$ database utilising all pre-2.3 Ga aged samples. SD = standard deviation, q# = the #th quantile.

Analysis Type	Count	Median	SD
SIMS	6479	0.138	1.900
Bulk SF ₆	2114	0.085	1.890
SHRIMP-SI	1191	0.078	1.170
Bulk EA-CF-IRMS	964	0.250	1.960
CO ₂ Spot Laser Fluorination, SF ₆	95	-0.049	0.534
Bulk MC-ICPMS	94	2.790	3.190
LA-MC-ICP-MS (spot)	91	0.390	0.509
In-situ Laser	21	-0.720	1.390
KrF Spot Fluorination, SF ₆	20	0.065	0.134

Table R4: Descriptive statistics of the pre-2.3 Ga database binned by analysis type. SD = standard deviation.

Baroni, M., Thiemens, M.H., Delmas, R.J., Savarino, J., 2007. Mass-Independent Sulfur Isotopic Compositions in Stratospheric Volcanic Eruptions. *Science* (80-.). 315, 84–87. <https://doi.org/10.1126/science.1131754>

Burke, A., Moore, K.A., Sigl, M., Nita, D.C., McConnell, J.R., Adkins, J.F., 2019. Stratospheric eruptions from tropical and extra-tropical volcanoes constrained using high-resolution sulfur isotopes in ice cores. *Earth Planet. Sci. Lett.* 521, 113–119. <https://doi.org/10.1016/j.epsl.2019.06.006>

Canfield, D.E., Farquhar, J., 2009. Animal evolution, bioturbation, and the sulfate concentration of the oceans. *Proc. Natl. Acad. Sci.* 106, 8123–8127. <https://doi.org/10.1073/pnas.0902037106>

- Fakhraee, M., Crowe, S.A., Katsev, S., 2018. Sedimentary sulfur isotopes and Neoproterozoic ocean oxygenation. *Sci. Adv.* 4, 1–6. <https://doi.org/10.1126/sciadv.1701835>
- Farquhar, J., Cliff, J., Zerkle, A.L., Kamyshny, A., Poulton, S.W., Claire, M., Adams, D., Harms, B., 2013. Pathways for Neoproterozoic pyrite formation constrained by mass-independent sulfur isotopes. *Proc. Natl. Acad. Sci. U. S. A.* 110, 17638–17643. <https://doi.org/10.1073/pnas.1218851110>
- Farquhar, J., Wing, B.A., 2003. Multiple sulfur isotopes and the evolution of the atmosphere. *Earth Planet. Sci. Lett.* 213, 1–13. [https://doi.org/10.1016/S0012-821X\(03\)00296-6](https://doi.org/10.1016/S0012-821X(03)00296-6)
- Gallagher, M., Whitehouse, M.J., Kamber, B.S., 2017. The Neoproterozoic surficial sulphur cycle: An alternative hypothesis based on analogies with 20th-century atmospheric lead. *Geobiology* 15, 385–400. <https://doi.org/10.1111/gbi.12234>
- Izon, G., Luo, G., Uveges, B.T., Beukes, N., Kitajima, K., Ono, S., Valley, J.W., Ma, X., Summons, R.E., 2022. Bulk and grain-scale minor sulfur isotope data reveal complexities in the dynamics of Earth's oxygenation. *Proc. Natl. Acad. Sci.* 119. <https://doi.org/10.1073/pnas.2025606119>
- Jamieson, J.W., Wing, B.A., Hannington, M.D., Farquhar, J., 2006. Evaluating isotopic equilibrium among sulfide mineral pairs in Archean ore deposits: case study from the Kidd Creek VMS deposit, Ontario, Canada. *Econ. Geol.* 101, 1055–1061. <https://doi.org/10.2113/gsecongeo.101.5.1055>
- Johnson, A.C., Ostrander, C.M., Romaniello, S.J., Reinhard, C.T., Greaney, A.T., Lyons, T.W., Anbar, A.D., 2021. Reconciling evidence of oxidative weathering and atmospheric anoxia on Archean Earth. *Sci. Adv.* 7, 1–10. <https://doi.org/10.1126/sciadv.abj0108>
- Johnson, J.E., Web, S.M., Thomas, K., Ono, S., Kirschvink, J.L., Fischer, W.W., 2013. Manganese-oxidizing photosynthesis before the rise of cyanobacteria. *Proc. Natl. Acad. Sci. U. S. A.* 110, 11238–11243. <https://doi.org/10.1073/pnas.1305530110>
- Junium, C.K., Zerkle, A.L., Witts, J.D., Ivany, L.C., Yancey, T.E., Liu, C., Claire, M.W., 2022. Massive perturbations to atmospheric sulfur in the aftermath of the Chicxulub impact. *Proc. Natl. Acad. Sci.* 119, 1–7. <https://doi.org/10.1073/pnas.2119194119>
- Luo, G., Ono, S., Beukes, N.J., Wang, D.T., Xie, S., Summons, R.E., 2016. Rapid oxygenation of Earth's atmosphere 2.33 billion years ago. *Sci. Adv.* 2, e1600134. <https://doi.org/10.1126/sciadv.1600134>
- Muller, É., Philippot, P., Rollion-Bard, C., Cartigny, P., Assayag, N., Marin-Carbonne, J., Mohan, M.R., Sarma, D.S., 2017. Primary sulfur isotope signatures preserved in high-grade Archean barite deposits of the Sargur Group, Dharwar Craton, India. *Precambrian Res.* 295, 38–47. <https://doi.org/10.1016/j.precamres.2017.04.029>
- Ono, S., Beukes, N.J., Rumble, D., 2009. Origin of two distinct multiple-sulfur isotope compositions of pyrite in the 2.5 Ga Klein Naute Formation, Griqualand West Basin, South Africa. *Precambrian Res.* 169, 48–57. <https://doi.org/10.1016/j.precamres.2008.10.012>
- Ono, S., Eigenbrode, J.L., Pavlov, A.A., Kharecha, P., Rumble, D., Kasting, J.F., Freeman, K.H., 2003. New insights into Archean sulfur cycle from mass-independent sulfur isotope records from the Hamersley Basin, Australia. *Earth Planet. Sci. Lett.* 213, 15–30. [https://doi.org/10.1016/S0012-821X\(03\)00295-4](https://doi.org/10.1016/S0012-821X(03)00295-4)
- Philippot, P., Ávila, J.N., Killingsworth, B.A., Tessalina, S., Baton, F., Caquineau, T., Muller, E., Pecoits, E., Cartigny, P., Lalonde, S. V., Ireland, T.R., Thomazo, C., van Kranendonk, M.J., Busigny, V., 2018. Globally asynchronous sulphur isotope signals require re-definition of the Great Oxidation Event. *Nat. Commun.* 9, 2245. <https://doi.org/10.1038/s41467-018-04621-x>
- Philippot, P., Van Zuilen, M., Rollion-Bard, C., 2012. Variations in atmospheric sulphur chemistry on early Earth linked to volcanic activity. *Nat. Geosci.* 5, 668–674. <https://doi.org/10.1038/ngeo1534>
- Planavsky, N.J., Bekker, A., Hofmann, A., Owens, J.D., Lyons, T.W., 2012. Sulfur record of rising and falling marine oxygen and sulfate levels during the Lomagundi event. *Proc. Natl. Acad. Sci. U. S. A.* 109, 18300–18305. <https://doi.org/10.1073/pnas.1120387109>
- Poulton, S.W., Bekker, A., Cumming, V.M., Zerkle, A.L., Canfield, D.E., Johnston, D.T., 2021. A 200-million-year delay in permanent atmospheric oxygenation. *Nature* 592, 232–236. <https://doi.org/10.1038/s41586-021-03393-7>
- Reinhard, C.T., Planavsky, N.J., Lyons, T.W., 2013. Long-term sedimentary recycling of rare sulphur isotope anomalies. *Nature* 497, 100–103. <https://doi.org/10.1038/nature12021>

- Ushikubo, T., Williford, K.H., Farquhar, J., Johnston, D.T., Van Kranendonk, M.J., Valley, J.W., 2014. Development of in situ sulfur four-isotope analysis with multiple Faraday cup detectors by SIMS and application to pyrite grains in a Paleoproterozoic glaciogenic sandstone. *Chem. Geol.* 383, 86–99.
<https://doi.org/10.1016/j.chemgeo.2014.06.006>
- Warke, M.R., Di Rocco, T., Zerkle, A.L., Lepland, A., Prave, A.R., Martin, A.P., Ueno, Y., Condon, D.J., Claire, M.W., 2020a. The Great Oxidation Event preceded a Paleoproterozoic “snowball Earth.” *Proc. Natl. Acad. Sci.* 117, 13314–13320. <https://doi.org/10.1073/pnas.2003090117>
- Warke, M.R., Strauss, H., Schröder, S., 2020b. Positive cerium anomalies imply pre-GOE redox stratification and manganese oxidation in Paleoproterozoic shallow marine environments. *Precambrian Res.* 344, 105767.
<https://doi.org/10.1016/j.precamres.2020.105767>
- Williford, K.H., Van Kranendonk, M.J., Ushikubo, T., Kozdon, R., Valley, J.W., 2011. Constraining atmospheric oxygen and seawater sulfate concentrations during Paleoproterozoic glaciation: In situ sulfur three-isotope microanalysis of pyrite from the Turee Creek Group, Western Australia. *Geochim. Cosmochim. Acta* 75, 5686–5705. <https://doi.org/10.1016/j.gca.2011.07.010>
- Wogan, N.F., Catling, D.C., Zahnle, K.J., Claire, M.W., 2022. Rapid timescale for an oxic transition during the Great Oxidation Event and the instability of low atmospheric O₂. *Proc. Natl. Acad. Sci.* 119, 2–9.
<https://doi.org/10.1073/pnas.2205618119>

Reviewer #3 (Remarks to the Author):

Dear authors, dear editor,

With a bit of a delay, please find below my comments and review of the new version of your manuscript.

Best regards

Guillaume Paris

In the second version of their manuscript submitted to Nature Communication, the authors made a very good job in improving the fluidity of the text and the quality of the explanations. I thank the authors for taking the time to answer my questions and the other reviewers's with great pedagogy.

I have a few additional remarks related to the new version of the text.

Mostly, I liked the conclusion of the previous version better. In the previous version of the text, stating the difference between the pre-Rooihogte possible oxidation events and post-Rooihogte "disoxygenation" events made the point clearer and stronger. Now, the conclusion appears diluted with the (interesting nonetheless) mentions of possible volcanic injections. I would suggest inverting the paragraphs line 213-225 and lines 226-247. Indeed, the notion of a fundamental transition is now less strong as the paragraph lines 226-247 tries to explain a way to not return to O₂-empoverished atmosphere and breaks to reasoning around the "fundamental transition" that is the heart of the paper, in my understanding.

I have two minor comments :

Line 15 : were instead of where ?

Line 50-51 : I realized I'm confused. I thought that you show (1) that the CME is much smaller than previously thought and that , as such, the seawater signal is much more coeval to atmospheric UV-SO₂ processes and (2) the positive spikes in D33S also reflect atmospheric changes with brief "deoxygenation" events. Maybe this sentence needs to be revisited in lights of the rest of the article?

Reviewer #4 (Remarks to the Author):

Review of Uveges et al., NatComm

Uveges et al., present a new analysis of S-MIF records across the critical interval of Earth's past where oxygen levels appear to have accumulated for the first time. The authors provide a detailed analysis of existing data and offer important new insights into how sampling bias, spatial bias and temporal bias may obfuscate the insights that the S-MIF offers. A focus of this work is evaluating recent results presented by Poulton et al., (2021, Nature), where a purported return of non zero S-MIF signatures are reported in a single sample set from S. Africa. In this work Poulton et al., took a strong stance that their results indicate a return to anoxic conditions after initial oxygenation. Uveges et al., rigorously interrogates this finding and conclude that either the community has under-sampled the sedimentary record between ≈ 2.4 -2.1 Ga and have been exceedingly unlucky to miss such intervals, or that additional work on the samples studied in Poulton et al., is needed to fully understand this resurgence in non-zero S-MIF.

I found this paper to be well written, very well laid out with clear and high-quality figures. I believe that this work is very important and will have a high impact on the Earth history community. Specifically, the dynamics surrounding Earth's oxygenation and what this can teach about the interplay of early life and the environment remains one of the central aims of fields such as geobiology. Results and analyses such as those presented by Uveges et al., are critical in that any meaningful inferences regarding early life must be grounded in an accurate representation of geochemical results from the sedimentary record. Beyond S-MIF, the analysis provided by Uveges

et al., offers an approach that I hope many will follow in interpreting many other geochemical results in the sedimentary record.

I have also reviewed the reviewer response document in detail and find that the authors have done a very good job of addressing previous concerns through the present manuscript.

My main suggestion focuses on the fact that many inferences are extended to the seawater sulfate reservoir (SWSR). But the authors do not appear to consider any existing sulfate $\Delta^{33}\text{S}$ data from the Paleoproterozoic sulfate record in the main text. Crockford et al., 2019 presented results from multiple Paleoproterozoic sulfates which all displayed near-zero $\Delta^{33}\text{S}$ values. While the relationship of any specific evaporite deposits barite deposit to contemporaneous seawater is debated, such results show a sharper transition to an oxygenated atmosphere than sulfide records given no apparent CME impacting this record. Moreover, such formations also appear to provide a more compelling case for the results of Torres et al., vs. Asael et al. and appear to favor a more homogenous SWSR across the post-oxygenation Paleoproterozoic. I feel it is important that these points are noted which I think could strengthen a number of points presented by the authors.

REVIEWERS' COMMENTS

Reviewer #3 (Remarks to the Author):

Dear authors, dear editor,

With a bit of a delay, please find below my comments and review of the new version of your manuscript.

Best regards

Guillaume Paris

In the second version of their manuscript submitted to Nature Communication, the authors made a very good job in improving the fluidity of the text and the quality of the explanations. I thank the authors for taking the time to answer my questions and the other reviewers's with great pedagogy.

We are glad that you found our revisions improved the manuscript, and we thank you again for your insight and helpful suggestions.

I have a few additional remarks related to the new version of the text. Mostly, I liked the conclusion of the previous version better. In the previous version of the text, stating the difference between the pre-Rooihogte possible oxidation events and post-Rooihogte "disoxygenation" events made the point clearer and stronger. Now, the conclusion appears diluted with the (interesting nonetheless) mentions of possible volcanic injections. I would suggest inverting the paragraphs line 213-225 and lines 226-247. Indeed, the notion of a fundamental transition is now less strong as the paragraph lines 226-247 tries to explain a way to not return to O₂-empoverished atmosphere and breaks to reasoning around the "fundamental transition" that is the heart of the paper, in my understanding.

Overall, we agree that the previous version did convey that particular conclusion of our work more clearly. However, at the behest of the other reviewers, we added in the discussion of two other competing models to explain the significance of the Timeball Hill S-MIF intervals. In our minds, the logical progression of discussing these alternate models was to discriminate between predominantly "oxic vs anoxic" background state in the TBH first, as that inherently determines the possibility of

short-term dynamics. Then, after establishing a more oxidizing prevailing state, we move on to discuss what could be causing the isolated occurrences of S-MIF in the TBH described by Poulton et al., starting with the volcanic models, and ending with our preferred interpretation. Aside from attempting to strike a balance between competing reviews, we feel that keeping the discussion/dismissal of the Plinian eruption models is warranted given that they are routinely suggested to us when discussing our work with the wider community. It is an idea that is obviously out there and has taken root, and therefore needs to be addressed in the context of our work. We have, however, attempted to smooth the transitions in the last section to make our thought process and conclusions clearer. We have also added the bit of text specifically delineating the difference between the pre-Rooihogte possible oxidation events and post-Rooihogte “deoxygenation” events back into the final paragraph at line 363. We hope that these changes amount to a suitable compromise.

I have two minor comments :

Line 15 : were instead of where ?

Thank you for the catch, we have fixed this typo

Line 50-51 : I realized I’m confused. I thought that you show (1) that the CME is much smaller than previously thought and that , as such, the seawater signal is much more coeval to atmospheric UV-SO₂ processes and (2) the positive spikes in D33S also reflect atmospheric changes with brief “deoxygenation” events. Maybe this sentence needs to be revisited in lights of the rest of the article?

We apologize for the confusion, and the poor wording on our part. In this instance “Here” was meant to refer the papers referenced in the previous sentence that DO argue for a large CME, rather than “here in this paper”. As you rightly point out, we conclude that the CME is overall small, and has the potential to be somewhat spatially variable. We have revised this portion of the text to be clearer (see lines 56-59 of the revised manuscript).

Reviewer #4 (Remarks to the Author):

Review of Uveges et al., NatComm

Uveges et al., present a new analysis of S-MIF records across the critical interval of Earth’s past where oxygen levels appear to have accumulated for the first time. The

authors provide a detailed analysis of existing data and offer important new insights into how sampling bias, spatial bias and temporal bias may obfuscate the insights that the S-MIF offers. A focus of this work is evaluating recent results presented by Poulton et al., (2021, Nature), where a purported return of non zero S-MIF signatures are reported in a single sample set from S. Africa. In this work Poulton et al., took a strong stance that their results indicate a return to anoxic conditions after initial oxygenation. Uveges et al., rigorously interrogates this finding and conclude that either the community has under-sampled the sedimentary record between ≈ 2.4 -2.1 Ga and have been exceedingly unlucky to miss such intervals, or that additional work on the samples studied in Poulton et al., is needed to fully understand this resurgence in non-zero S-MIF.

We thank you for an excellent precis of our paper and are very encouraged that this particular message came through in our discussion!

I found this paper to be well written, very well laid out with clear and high-quality figures. I believe that this work is very important and will have a high impact on the Earth history community. Specifically, the dynamics surrounding Earth's oxygenation and what this can teach about the interplay of early life and the environment remains one of the central aims of fields such as geobiology. Results and analyses such as those presented by Uveges et al., are critical in that any meaningful inferences regarding early life must be grounded in an accurate representation of geochemical results from the sedimentary record. Beyond S-MIF, the analysis provided by Uveges et al., offers an approach that I hope many will follow in interpreting many other geochemical results in the sedimentary record.

We thank you for such supportive comments and are very happy you found the paper clear and impactful.

I have also reviewed the reviewer response document in detail and find that the authors have done a very good job of addressing previous concerns through the present manuscript.

My main suggestion focuses on the fact that many inferences are extended to the seawater sulfate reservoir (SWSR). But the authors do not appear to consider any existing sulfate $\Delta 33S$ data from the Paleoproterozoic sulfate record in the main text. Crockford et al., 2019 presented results from multiple Paleoproterozoic sulfates which all displayed near-zero $\Delta 33S$ values. While the relationship of any specific evaporite deposits barite deposit to contemporaneous seawater is debated, such

results show a sharper transition to an oxygenated atmosphere than sulfide records given no apparent CME impacting this record. Moreover, such formations also appear to provide a more compelling case for the results of Torres et al., vs. Asael et al. and appear to favor a more homogenous SWSR across the post-oxygenation Paleoproterozoic. I feel it is important that these points are noted which I think could strengthen a number of points presented by the authors.

This is an excellent point, and one that we agree needed to be worked into the text. We have added a few lines to the main text (line 147) to make reference to Crockford et al., 2019 and other $\Delta^{33}\text{S}$ values in post 2.3Ga sulphates. As the reviewer rightly suggests, this does strengthen the case for a negligible $\Delta^{33}\text{S}$ value in the globally integrated SWSR. This comment also reminded us of the fact that Killingsworth et al. (2018) reported $\Delta^{33}\text{S}$ data from barites in the Kazput Formation of the Turee Creek Group. These sulphate minerals should ostensibly record the composition of the seawater sulphate reservoir in the TCG basin, and they track very closely with cooccurring pyrite-derived $\Delta^{33}\text{S}$ values for the Kazput. This lends further credence to our proposed model of a somewhat spatially variable SWSR $\Delta^{33}\text{S}$ composition. We have added a discussion of these data in relation to those presented in Crockford et al., 2019 (and references therein) to the text at line 202 and have modified figure 1e to show the data of Killingsworth et al. superimposed on those of Philippot et al.